# HETEROGENEOUS-AGENT MIRROR LEARNING

## ABSTRACT

The necessity for cooperation among independent intelligent machines has popularised cooperative *multi-agent reinforcement learning* (MARL) in the artificial intelligence (AI) research community. However, many research endeavours have been focused on developing practical MARL algorithms whose effectiveness has been studied only empirically, thereby lacking theoretical guarantees. As recent studies have revealed, MARL methods often achieve performance that is unstable in terms of reward monotonicity or suboptimal at convergence. To resolve these issues, in this paper,we introduce a novel framework named *Heterogeneous-Agent Mirror Learning* (HAML) that provides a general template for MARL actor-critic algorithms. We prove that algorithms derived from the HAML template satisfy the desired properties of the monotonic improvement of the joint reward and the convergence to *Nash equilibrium*. We verify the practicality of HAML by proving that the current state-of-the-art cooperative MARL algorithms, HATRPO and HAPPO, are in fact HAML instances. Next, as a natural outcome of our theory, we propose HAML extensions of two well-known RL algorithms, HAA2C (for A2C) and HADDPG (for DDPG), and demonstrate their effectiveness against strong baselines on StarCraftII and Multi-Agent MuJoCo tasks.

## 1 INTRODUCTION

While the policy gradient (PG) formula has been long known in the *reinforcement learning* (RL) community (Sutton et al., 2000), it has not been until *trust region learning* (Schulman et al., 2015a) that deep RL algorithms started to solve complex tasks such as real-world robotic control successfully. Nowadays, methods that followed the trust-region framework, including TRPO (Schulman et al., 2015a), PPO (Schulman et al., 2017) and their extensions (Schulman et al., 2015b; Hsu et al., 2020), became effective tools for solving challenging AI problems (Berner et al., 2019). It was believed that the key to their success are the rigorously described stability and the monotonic improvement property of trust-region learning that they approximate. This reasoning, however, would have been of limited scope since it failed to explain why some algorithms following it (*e.g.* PPO-KL) largely underperform in contrast to success of other ones (*e.g.* PPO-clip) (Schulman et al., 2017). Furthermore, the trust-region interpretation of PPO has been formally rejected by recent studies both empirically (Engstrom et al., 2020) and theoretically (Wang et al., 2020); this revealed that the algorithm violates the trust-region constraints—it neither constraints the KL-divergence between two consecutive policies, nor does it bound their likelihood ratios. These findings have suggested that, while the number of available RL algorithms grows, our understanding of them does not, and the algorithms often come without theoretical guarantees either.

Only recently, Kuba et al. (2022b) showed that the well-known algorithms, such as PPO, are in fact instances of the so-called *mirror learning* framework, within which any induced algorithm is theoretically sound. On a high level, methods that fall into this class optimise the *mirror objective*, which shapes an advantage surrogate by means of a *drift functional*—a quasi-distance between policies. Such an update provably leads them to monotonic improvements of the return, as well as the convergence to the optimal policy. The result of mirror learning offers RL researchers strong confidence that there exists a connection between an algorithm's practicality and its theoretical properties and assures soundness of the common RL practice.

While the problem of the lack of theoretical guarantees has been severe in RL, in *multi-agent reinforcement learning* (MARL) it has only been exacerbated. Although the PG theorem has been successfully extended to the multi-agent PG (MAPG) version (Zhang et al., 2018), it has only recently

been shown that the variance of MAPG estimators grows linearly with the number of agents (Kuba et al., 2021). Prior to this, however, a novel paradigm of *centralised training for decentralised execution* (CTDE) (Foerster et al., 2018; Lowe et al., 2017b) greatly alleviated the difficulty of the multi-agent learning by assuming that the global state and opponents' actions and policies are accessible during the training phase; this enabled developments of practical MARL methods by merely extending single-agent algorithms' implementations to the multi-agent setting. As a result, direct extensions of TRPO (Li & He, 2020) and PPO (de Witt et al., 2020a; Yu et al., 2021) have been proposed whose performance, although is impressive in some settings, varies according to the version used and environment tested against. However, these extensions do not assure the monotonic improvement property or convergence result of any kind (Kuba et al., 2022a). Importantly, these methods can be proved to be suboptimal at convergence in the common setting of *parameter sharing* (Kuba et al., 2022a) which is considered as default by popular multi-agent algorithms (Yu et al., 2021) and popular multi-agent benchmarks such as SMAC (Samvelyan et al., 2019) due to the computational convenience it provides.

In this paper, we resolve these issues by proposing *Heterogeneous-Agent Mirror Learning* (HAML)— a template that can induce a continuum of cooperative MARL algorithms with theoretical guarantees for monotonic improvement as well as *Nash equilibrium* (NE) convergence. The purpose of HAML is to endow MARL researchers with a template for rigorous algorithmic design so that having been granted a method's correctness upfront, they can focus on other aspects, such as effective implementation through deep neural networks. We demonstrate the expressive power of the HAML framework by showing that two of existing state-of-the-art (SOTA) MARL algorithms, HATRPO and HAPPO (Kuba et al., 2022a), are rigorous instances of HAML. This stands in contrast to viewing them as merely approximations to provably correct multi-agent trust-region algorithms as which they were originally considered. Furthermore, although HAML is mainly a theoretical contribution, we can naturally demonstrate its usefulness by using it to derive two heterogeneous-agent extensions of successful RL algorithms: HAA2C (for A2C (Mnih et al., 2016)) and HADDPG (for DDPG (Lillicrap et al., 2015)), whose strength are demonstrated on benchmarks of StarCraftII (SMAC) (de Witt et al., 2020a) and Multi-Agent MuJoCo (de Witt et al., 2020b) against strong baselines such as MADDPG (Lowe et al., 2017b) and MAA2C (Papoudakis et al., 2021).

## 2 PROBLEM FORMULATION

We formulate the cooperative MARL problem as *cooperative Markov game* (Littman, 1994) defined by a tuple $\langle \mathcal{N}, \mathcal{S}, \boldsymbol{\mathcal{A}}, r, P, \gamma, d \rangle$. Here, $\mathcal{N} = \{1, \ldots, n\}$ is a set of $n$ agents, $\mathcal{S}$ is the state space, $\boldsymbol{\mathcal{A}} = \times_{i=1}^{n} \mathcal{A}^i$ is the products of all agents' action spaces, known as the joint action space. Although our results hold for general compact state and action spaces, in this paper we assume that they are finite, for simplicity. Further, $r : \mathcal{S} \times \boldsymbol{\mathcal{A}} \to \mathbb{R}$ is the joint reward function, $P : \mathcal{S} \times \boldsymbol{\mathcal{A}} \times \mathcal{S} \to [0, 1]$ is the transition probability kernel, $\gamma \in [0, 1)$ is the discount factor, and $d \in \mathcal{P}(\mathcal{S})$ (where $\mathcal{P}(X)$ denotes the set of probability distributions over a set $X$) is the positive initial state distribution. In this work, we will also use the notation $\mathbb{P}(X)$ to denote the power set of a set $X$. At time step $t \in \mathbb{N}$, the agents are at state $\mathrm{s}_t$ (which may not be fully observable); they take independent actions $\mathrm{a}_t^i, \forall i \in \mathcal{N}$ drawn from their policies $\pi^i(\cdot^i|\mathrm{s}_t) \in \mathcal{P}(\mathcal{A}^i)$, and equivalently, they take a joint action $\mathbf{a}_t = (\mathrm{a}_t^1, \ldots, \mathrm{a}_t^n)$ drawn from their joint policy $\boldsymbol{\pi}(\cdot|\mathrm{s}_t) = \prod_{i=1}^{n} \pi^i(\cdot^i|\mathrm{s}_t) \in \mathcal{P}(\boldsymbol{\mathcal{A}})$. We write $\Pi^i \triangleq \{\times_{s \in \mathcal{S}} \pi^i(\cdot^i|s) \,|\forall s \in \mathcal{S}, \pi^i(\cdot^i|s) \in \mathcal{P}(\mathcal{A}^i)\}$ to denote the policy space of agent $i$, and $\boldsymbol{\Pi} \triangleq (\Pi^1, \ldots, \Pi^n)$ to denote the joint policy space. It is important to note that when $\pi^i(\cdot^i|s)$ is a Dirac delta ditribution, the policy is referred to as *deterministic* (Silver et al., 2014) and we write $\mu^i(s)$ to refer to its centre. Then, the environment emits the joint reward $r(\mathrm{s}_t, \mathbf{a}_t)$ and moves to the next state $\mathrm{s}_{t+1} \sim P(\cdot|\mathrm{s}_t, \mathbf{a}_t) \in \mathcal{P}(\mathcal{S})$. The initial state distribution $d$, the joint policy $\boldsymbol{\pi}$, and the transition kernel $P$ induce the (improper) marginal state distribution $\rho_{\boldsymbol{\pi}}(s) \triangleq \sum_{t=0}^{\infty} \gamma^t \mathrm{Pr}(\mathrm{s}_t = s|d, \boldsymbol{\pi})$. The agents aim to maximise the expected joint return, defined as

$$J(\boldsymbol{\pi}) = \mathbb{E}_{\mathrm{s}_0 \sim d, \mathbf{a}_{0:\infty} \sim \boldsymbol{\pi}, \mathrm{s}_{1:\infty} \sim P} \Big[ \sum_{t=0}^{\infty} \gamma^t r(\mathrm{s}_t, \mathbf{a}_t) \Big].$$

We adopt the most common solution concept for multi-agent problems which is that of *Nash equilibria* (Nash, 1951; Yang & Wang, 2020). We say that a joint policy $\boldsymbol{\pi}_{\mathrm{NE}} \in \boldsymbol{\Pi}$ is a NE if none of the agents can increase the joint return by unilaterally altering its policy. More formally, $\boldsymbol{\pi}_{\mathrm{NE}}$ is a NE if

$$\forall i \in \mathcal{N}, \forall \pi^i \in \Pi^i, \ J(\pi^i, \boldsymbol{\pi}_{\mathrm{NE}}^{-i}) \leq J(\boldsymbol{\pi}_{\mathrm{NE}}).$$

To study the problem of finding a NE, we introduce the following notions. Let $i_{1:m} = (i_1, \ldots, i_m) \subseteq \mathcal{N}$ be an ordered subset of agents. We write $-i_{1:m}$ to refer to its complement, and $i$ and $-i$, respectively, when $m = 1$. We define the multi-agent state-action value function as

$$Q_{\boldsymbol{\pi}}^{i_{1:m}}(s, \boldsymbol{a}^{i_{1:m}}) \triangleq \mathbb{E}_{\mathbf{a}_0^{-i_{1:m}} \sim \boldsymbol{\pi}^{-i_{1:m}}, s_{1:\infty} \sim P, \mathbf{a}_{1:\infty} \sim \boldsymbol{\pi}} \Big[ \sum_{t=0}^{\infty} \gamma^t r(s_t, \mathbf{a}_t) \,\Big|\, s_0 = s, \mathbf{a}_0^{i_{1:m}} = \boldsymbol{a}^{i_{1:m}} \Big].$$

When $m = n$ (the joint action of all agents is considered), then $i_{1:n} \in \mathrm{Sym}(n)$, where $\mathrm{Sym}(n)$ denotes the set of permutations of integers $1, \ldots, n$, known as the ***symmetric group***. In that case we write $Q_{\boldsymbol{\pi}}^{i_{1:n}}(s, \boldsymbol{a}^{i_{1:n}}) = Q_{\boldsymbol{\pi}}(s, \boldsymbol{a})$ which is known as the (joint) state-action value function. On the other hand, when $m = 0$, *i.e.*, $i_{1:m} = \emptyset$, the function takes the form $V_{\boldsymbol{\pi}}(s)$, known as the state-value function. Consider two disjoint subsets of agents, $j_{1:k}$ and $i_{1:m}$. Then, the multi-agent advantage function of $i_{1:m}$ with respect to $j_{1:k}$ is defined as

$$A_{\boldsymbol{\pi}}^{i_{1:m}}(s, \boldsymbol{a}^{j_{1:k}}, \boldsymbol{a}^{i_{1:m}}) \triangleq Q_{\boldsymbol{\pi}}^{j_{1:k}, i_{1:m}}(s, \boldsymbol{a}^{j_{1:k}}, \boldsymbol{a}^{i_{1:m}}) - Q_{\boldsymbol{\pi}}^{j_{1:k}}(s, \boldsymbol{a}^{j_{1:k}}).$$

As it has been shown in Kuba et al. (2021), the multi-agent advantage function allows for additive decomposition of the joint advantge function by the means of the following lemma.

**Lemma 1** (Multi-Agent Advantage Decomposition). *Let $\boldsymbol{\pi}$ be a joint policy, and $i_1, \ldots, i_m$ be an arbitrary ordered subset of agents. Then, for any state $s$ and joint action $\boldsymbol{a}^{i_{1:m}}$,*

$$A_{\boldsymbol{\pi}}^{i_{1:m}}(s, \boldsymbol{a}^{i_{1:m}}) = \sum_{j=1}^{m} A_{\boldsymbol{\pi}}^{i_j}(s, \boldsymbol{a}^{i_{1:j-1}}, a^{i_j}). \tag{1}$$

Although the multi-agent advantage function has been discovered only recently and has not been studied thoroughly, it is this function and the above lemma that builds the foundation for the development of the HAML theory.

## 3  THE STATE OF AFFAIRS IN MARL

Before we review existing SOTA algorithms for cooperative MARL, we introduce two settings in which the algorithms can be implemented. Both of them can be considered appealing depending on the application, but these pros also come with "traps" which, if not taken care of, may provably deteriorate an algorithm's performance.

### 3.1  HOMOGENEITY *vs.* HETEROGENEITY

The first setting is that of *homogeneous* policies, *i.e.*, those where agents all agents share one policy: $\pi^i = \pi, \forall \mathcal{N}$, so that $\boldsymbol{\pi} = (\pi, \ldots, \pi)$ (de Witt et al., 2020a; Yu et al., 2021). This approach enables a straightforward adoption of an RL algorithm to MARL, and it does not introduce much computational and sample complexity burden with the increasing number of agents. Nevertheless, sharing one policy across all agents requires that their action spaces are also the same, *i.e.*, $\mathcal{A}^i = \mathcal{A}^j, \forall i, j \in \mathcal{N}$. Furthermore, policy sharing prevents agents from learning different skills. This scenario may be particularly dangerous in games with a large number of agents, as shown in Proposition 1, proved by Kuba et al. (2022a).

**Proposition 1** (Trap of Homogeneity). *Let's consider a fully-cooperative game with an even number of agents $n$, one state, and the joint action space $\{0, 1\}^n$, where the reward is given by $r(\mathbf{0}^{n/2}, \mathbf{1}^{n/2}) = r(\mathbf{1}^{n/2}, \mathbf{0}^{n/2}) = 1$, and $r(\boldsymbol{a}^{1:n}) = 0$ for all other joint actions. Let $J^*$ be the optimal joint reward, and $J^*_{share}$ be the optimal joint reward under the shared policy constraint. Then*

$$\frac{J^*_{share}}{J^*} = \frac{2}{2^n}.$$

A more ambitious approach to MARL is to allow for *heterogeneity* of policies among agents, *i.e.*, to let $\pi^i$ and $\pi^j$ be different functions when $i \neq j \in \mathcal{N}$. This setting has greater applicability as heterogeneous agents can operate in different action spaces. Furthermore, thanks to this model's flexibility they may learn more sophisticated joint behaviours. Lastly, they can recover homogeneous policies as a result of training, if that is indeed optimal. Nevertheless, training heterogeneous agents is highly non-trivial. Given a joint reward, an individual agent may not be able to distill its own contibution to it—a problem known as *credit assignment* (Foerster et al., 2018). Furthermore, even if an agent identifies its improvement direction, it may be conflicting with those of other agents, which then results in performance damage, as we exemplify in Proposition 2, proved in Appendix A.2.

**Proposition 2** (Trap of Heterogeneity). *Let's consider a fully-cooperative game with* $2$ *agents, one state, and the joint action space* $\{0,1\}^2$, *where the reward is given by* $r(0,0) = 0, r(0,1) = r(1,0) = 2$, *and* $r(1,1) = -1$. *Suppose that* $\pi_{old}^i(0) > 0.6$ *for* $i = 1, 2$. *Then, if agents* $i$ *update their policies by*

$$\pi_{new}^i = \arg\max_{\pi^i} \mathbb{E}_{a^i \sim \pi^i, a^{-i} \sim \pi_{old}^{-i}}\left[A_{\boldsymbol{\pi}_{old}}(a^i, a^{-i})\right], \forall i \in \mathcal{N},$$

*then the resulting policy will yield a lower return,*

$$J(\boldsymbol{\pi}_{old}) > J(\boldsymbol{\pi}_{new}) = \min_{\boldsymbol{\pi}} J(\boldsymbol{\pi}).$$

Consequently, these facts imply that homogeneous algorithms should not be applied to complex problems, but they also highlight that heterogeneous algorithms should be developed with extra cares. In the next subsection, we describe existing SOTA actor-critic algorithms which, while often performing greatly, are still not impeccable, as they fall into one of the above two traps.

### 3.2 A Second Look at SOTA MARL Algorithms

**Multi-Agent Advantage Actor-Critic**    MAA2C (Papoudakis et al., 2021) extends the A2C (Mnih et al., 2016) algorithm to MARL by replacing the RL optimisation (single-agent policy) objective with the MARL one (joint policy),

$$\mathcal{L}^{\text{MAA2C}}(\boldsymbol{\pi}) \triangleq \mathbb{E}_{s \sim \boldsymbol{\pi}, a \sim \boldsymbol{\pi}}\left[A_{\boldsymbol{\pi}_{\text{old}}}(s, a)\right], \tag{2}$$

which computes the gradient with respect to every agent $i$'s policy parameters, and performs a gradient-ascent update for each agent. This algorithm is straightforward to implement and is capable of solving simple multi-agent problems (Papoudakis et al., 2021). We point out, however, that by simply following their own MAPG, the agents perform uncoordinated updates, thus getting caught by Proposition 2. Furthermore, MAPG estimates suffer from large variance which grows linearly with the number of agents (Kuba et al., 2021), thus making the algorithm unstable. To assure greater stability, the following MARL extensions of stable RL approaches have been developed.

**Multi-Agent Deep Deterministic Policy Gradient**    MADDPG (Lowe et al., 2017a) is a MARL extension of the popular DDPG algorithm (Lillicrap et al., 2015). At every iteration, every agent $i$ updates its deterministic policy by maximising the following objective

$$\mathcal{L}_i^{\text{MADDPG}}(\mu^i) \triangleq \mathbb{E}_{s \sim \beta_{\boldsymbol{\mu}_{\text{old}}}}\left[Q_{\boldsymbol{\mu}_{\text{old}}}^i\left(s, \mu^i(s)\right)\right] = \mathbb{E}_{s \sim \beta_{\boldsymbol{\mu}_{\text{old}}}}\left[Q_{\boldsymbol{\mu}_{\text{old}}}^i\left(s, \mu^i, \boldsymbol{\mu}_{\text{old}}^{-i}(s)\right)\right], \tag{3}$$

where $\beta_{\boldsymbol{\mu}_{\text{old}}}$ is a state distribution that is not necessarily equivalent to $\rho_{\boldsymbol{\mu}_{\text{old}}}$, thus allowing for off-policy training. In practice, MADDPG maximises Equation (3) by a few steps of gradient ascent. The main advantages of MADDPG include small variance of its MAPG estimates—a property granted by deterministic policies (Silver et al., 2014), as well as low sample complexity due to learning from off-policy data. Such a combination gives the algorithm a strong performance in continuous-action tasks (Lowe et al., 2017a; Kuba et al., 2022a). However, this method's strengths also constitute its limitations—it is applicable to continuous problems only (discrete problems require categorical-distribution policies), and relies on large memory capacity to store the off-policy data.

**Multi-Agent PPO**    MAPPO (Yu et al., 2021) is a relatively straightforward extension of PPO (Schulman et al., 2017) to MARL. In its default formulation, the agents employ the trick of *policy sharing* described in the previous subsection. As such, the policy is updated to maximise

$$\mathcal{L}^{\text{MAPPO}}(\boldsymbol{\pi}) \triangleq \mathbb{E}_{s \sim \rho_{\boldsymbol{\pi}_{\text{old}}}, a \sim \boldsymbol{\pi}_{\text{old}}}\left[\sum_{i=1}^n \min\left(\frac{\pi(a^i|s)}{\pi_{\text{old}}(a^i|s)}A_{\boldsymbol{\pi}_{\text{old}}}(s, a), \text{clip}\left(\frac{\pi(a^i|s)}{\pi_{\text{old}}(a^i|s)}, 1 \pm \epsilon\right)A_{\boldsymbol{\pi}_{\text{old}}}(s, a)\right)\right], \tag{4}$$

where the clip$(\cdot, 1 \pm \epsilon)$ operator clips the input to $1 - \epsilon/1 + \epsilon$ if it is below/above this value. Such an operation removes the incentive for agents to make large policy updates, thus stabilising the training effectively. Indeed, the algorithm's performance on the StarCraftII benchmark is remarkable, and it is accomplished by using only on-policy data. Nevertheless, the policy-sharing strategy limits the algorithm's applicability and leads to its suboptimality, as we discussed in Proposition 1. Trying to

avoid this issue, one can implement the algorithm without policy sharing, thus making the agents simply take simultaneous PPO updates meanwhile employing a joint advantage estimator. In this case, the updates are not coordinated, making MAPPO fall into the trap of Proposition 2.

In summary, all these algorithms do not possess performance guarantees. Altering their implementation settings to escape one of the traps from Subsection 3.1 makes them, at best, fall into another. This shows that the MARL problem introduces additional complexity into the single-agent RL setting, and needs additional care to be rigorously solved. With this motivation, in the next section, we develop a theoretical framework for development of MARL algorithms with correctness guarantees.

## 4 HETEROGENEOUS-AGENT MIRROR LEARNING

In this section, we introduce *heterogeneous-agent mirror learning* (HAML)—a template that includes a continuum of MARL algorithms which we prove to solve cooperative problems with correctness guarantees. HAML is designed for the general and expressive setting of heterogeneous agents, thus avoiding Proposition 1, and it is capable of coordinating their updates, leaving Proposition 2 behind.

### 4.1 SETTING UP HAML

We begin by introducing the necessary definitions of HAML attributes: the drift functional.

**Definition 1.** *Let $i \in \mathcal{N}$, a **heterogeneous-agent drift functional** (HADF) $\mathfrak{D}^{i,\nu}$ of $i$ consists of a map, which is defined as*

$$\mathfrak{D}^i : \mathbf{\Pi} \times \mathbf{\Pi} \times \mathbb{P}(-i) \times \mathcal{S} \to \{\mathfrak{D}^i_{\boldsymbol{\pi}}(\cdot|s, \bar{\boldsymbol{\pi}}^{j_{1:m}}) : \mathcal{P}(\mathcal{A}^i) \to \mathbb{R}\},$$

*such that for all arguments, under notation $\mathfrak{D}^i_{\boldsymbol{\pi}}(\hat{\pi}^i|s, \bar{\boldsymbol{\pi}}^{j_{1:m}}) \triangleq \mathfrak{D}^i_{\boldsymbol{\pi}}(\hat{\pi}^i(\cdot^i|s)|\bar{\boldsymbol{\pi}}^{j_{1:m}}, s),$*

1. $\mathfrak{D}^i_{\boldsymbol{\pi}}(\hat{\pi}^i|s, \bar{\boldsymbol{\pi}}^{j_{1:m}}) \geq \mathfrak{D}^i_{\boldsymbol{\pi}}(\pi^i|s, \bar{\boldsymbol{\pi}}^{j_{1:m}}) = 0$ *(non-negativity),*

2. $\mathfrak{D}^i_{\boldsymbol{\pi}}(\hat{\pi}^i|s, \bar{\boldsymbol{\pi}}^{j_{1:m}})$ *has all Gâteaux derivatives zero at $\hat{\pi}^i = \pi^i$ (zero gradient),*

*and a probability distribution $\nu^i_{\boldsymbol{\pi},\hat{\pi}^i} \in \mathcal{P}(\mathcal{S})$ over states that can (but does not have to) depend on $\boldsymbol{\pi}$ and $\hat{\pi}^i$, and such that the drift $\mathfrak{D}^{i,\nu}$ of $\hat{\pi}^i$ from $\pi^i$ with respect to $\bar{\boldsymbol{\pi}}^{j_{1:m}}$, defined as*

$$\mathfrak{D}^{i,\nu}_{\boldsymbol{\pi}}(\hat{\pi}^i|\bar{\boldsymbol{\pi}}^{j_{1:m}}) \triangleq \mathbb{E}_{\mathbf{s}\sim\nu^i_{\boldsymbol{\pi},\hat{\pi}^i}}\big[\mathfrak{D}^i_{\boldsymbol{\pi}}(\hat{\pi}^i|\mathbf{s}, \bar{\boldsymbol{\pi}}^{j_{1:m}})\big],$$

*is continuous with respect to $\boldsymbol{\pi}, \bar{\boldsymbol{\pi}}^{j_{1:m}}$, and $\hat{\pi}^i$. We say that the HADF is positive if $\mathfrak{D}^{i,\nu}_{\boldsymbol{\pi}}(\hat{\pi}^i|\bar{\boldsymbol{\pi}}^{j_{1:m}}) = 0$ implies $\hat{\pi}^i = \pi^i$, and trivial if $\mathfrak{D}^{i,\nu}_{\boldsymbol{\pi}}(\hat{\pi}^i|\bar{\boldsymbol{\pi}}^{j_{1:m}}) = 0$ for all $\boldsymbol{\pi}, \bar{\boldsymbol{\pi}}^{j_{1:m}}$, and $\hat{\pi}^i$, .*

Intuitively, the drift $\mathfrak{D}^{i,\nu}_{\boldsymbol{\pi}}(\hat{\pi}^i|\bar{\boldsymbol{\pi}}^{j_{1:m}})$ is a notion of distance between $\pi^i$ and $\hat{\pi}^i$, given that agents $j_{1:m}$ just updated to $\bar{\boldsymbol{\pi}}^{j_{1:m}}$. We highlight that, under this conditionality, the same update (from $\pi^i$ to $\hat{\pi}^i$) can have different sizes—this will later enable HAML agents to *softly* constraint their learning steps in a coordinated way. Before that, we introduce a notion that renders *hard* constraints, which may be a part of an algorithm design, or an inherrent limitation.

**Definition 2.** *Let $i \in \mathcal{N}$. We say that, $\mathcal{U}^i : \mathbf{\Pi} \times \Pi^i \to \mathbb{P}(\Pi^i)$ is a neighbourhood operator if $\forall \pi^i \in \Pi^i$, $\mathcal{U}^i_{\boldsymbol{\pi}}(\pi^i)$ contains a closed ball, i.e., there exists a state-wise monotonically non-decreasing metric $\chi : \Pi^i \times \Pi^i \to \mathbb{R}$ such that $\forall \pi^i \in \Pi^i$ there exists $\delta^i > 0$ such that $\chi(\pi^i, \bar{\pi}^i) \leq \delta^i \implies \bar{\pi}^i \in \mathcal{U}^i_{\boldsymbol{\pi}}(\pi^i)$.*

Throughout this work, for every joint policy $\boldsymbol{\pi}$, we will associate it with its *sampling distribution*—a positive state distribution $\beta_{\boldsymbol{\pi}} \in \mathcal{P}(\mathcal{S})$ that is continuous in $\boldsymbol{\pi}$ (Kuba et al., 2022b). With these notions defined, we introduce the main definition of the paper.

**Definition 3.** *Let $i \in \mathcal{N}$, $j^{1:m} \in \mathbb{P}(-i)$, and $\mathfrak{D}^{i,\nu}$ be a HADF of agent $i$. The **heterogeneous-agent mirror operator** (HAMO) integrates the advantage function as*

$$\big[\mathcal{M}^{(\hat{\pi}^i)}_{\mathfrak{D}^{i,\nu}, \bar{\boldsymbol{\pi}}^{j_{1:m}}} A_{\boldsymbol{\pi}}\big](s) \triangleq \mathbb{E}_{\mathbf{a}^{j_{1:m}}\sim\bar{\boldsymbol{\pi}}^{j_{1:m}}, \mathbf{a}^i\sim\hat{\pi}^i}\big[A^i_{\boldsymbol{\pi}}(s, \mathbf{a}^{j_{1:m}}, \mathbf{a}^i)\big] - \frac{\nu^i_{\boldsymbol{\pi},\hat{\pi}^i}(s)}{\beta_{\boldsymbol{\pi}}(s)}\mathfrak{D}^i_{\boldsymbol{\pi}}(\hat{\pi}^i|s, \bar{\boldsymbol{\pi}}^{j_{1:m}}).$$

We note two important facts. First, when $\nu^i_{\boldsymbol{\pi},\hat{\pi}^i} = \beta_{\boldsymbol{\pi}}$, the fraction from the front of the HADF in HAMO disappears, making it only a difference between the advantage and the HADF's evaluation. Second, when $\hat{\pi}^i = \pi^i$, HAMO evaluates to zero. Therefore, as the HADF is non-negative, a policy $\hat{\pi}^i$ that improves HAMO must make it positive, and thus leads to the improvement of the multi-agent advantage of agent $i$. In the next subsection, we study the properties of HAMO in more details, as well as use it to construct HAML—a general framework for MARL algorithm design.

### 4.2 Theoretical Properties of HAML

It turns out that, under certain configuration, agents' local improvements result in the joint improvement of all agents, as described by the lemma below.

**Lemma 2** (HAMO Is All You Need). *Let $\boldsymbol{\pi}_{old}$ and $\boldsymbol{\pi}_{new}$ be joint policies and let $i_{1:n} \in \mathrm{Sym}(n)$ be an agent permutation. Suppose that, for every state $s \in \mathcal{S}$ and every $m = 1, \dots, n$,*

$$\big[\mathcal{M}^{(\pi_{new}^{i_m})}_{\mathfrak{D}^{i_m,\nu}, \boldsymbol{\pi}_{new}^{i_{1:m-1}}} A_{\boldsymbol{\pi}_{old}}\big](s) \geq \big[\mathcal{M}^{(\pi_{old}^{i_m})}_{\mathfrak{D}^{i_m,\nu}, \boldsymbol{\pi}_{new}^{i_{1:m-1}}} A_{\boldsymbol{\pi}_{old}}\big](s). \tag{5}$$

*Then, $\boldsymbol{\pi}_{new}$ is jointly better than $\boldsymbol{\pi}_{old}$, so that for every state $s$,*

$$V_{\boldsymbol{\pi}_{new}}(s) \geq V_{\boldsymbol{\pi}_{old}}(s).$$

Subsequently, the *monotonic improvement property* of the joint return follows naturally, as

$$J(\boldsymbol{\pi}_{\text{new}}) = \mathbb{E}_{s\sim d}\big[V_{\boldsymbol{\pi}_{\text{new}}}(s)\big] \geq \mathbb{E}_{s\sim d}\big[V_{\boldsymbol{\pi}_{\text{old}}}(s)\big] = J(\boldsymbol{\pi}_{\text{old}}).$$

However, the conditions of the lemma require every agent to solve $|\mathcal{S}|$ instances of Inequality (5), which may be an intractable problem. We shall design a single optimisation objective whose solution satisfies those inequalities instead. Furthermore, to have a practical application to large-scale problems, such an objective should be estimatable via sampling. To handle these challenges, we introduce the following Algorithm Template 1 which generates a continuum of HAML algorithms.

---

**Algorithm Template 1:** Heterogeneous-Agent Mirror Learning

---

Initialise a joint policy $\boldsymbol{\pi}_0 = (\pi_0^1, \dots, \pi_0^n)$;

**for** $k = 0, 1, \dots$ **do**

    Compute the advantage function $A_{\boldsymbol{\pi}_k}(s, \boldsymbol{a})$ for all state-(joint)action pairs $(s, \boldsymbol{a})$;

    Draw a permutaion $i_{1:n}$ of agents at random \\from a positive distribution $p \in \mathcal{P}(\mathrm{Sym}(n))$;

    **for** $m = 1 : n$ **do**

        Make an update $\pi_{k+1}^{i_m} = \underset{\pi^{i_m} \in \mathcal{U}_{\boldsymbol{\pi}_k}^{i_m}(\pi_k^{i_m})}{\arg\max} \mathbb{E}_{s\sim\beta_{\boldsymbol{\pi}_k}} \Big[ \big[\mathcal{M}^{(\pi^{i_m})}_{\mathfrak{D}^{i_m,\nu}, \boldsymbol{\pi}_{k+1}^{i_{1:m-1}}} A_{\boldsymbol{\pi}_k}\big](s) \Big]$;

**Output :** A limit-point joint policy $\boldsymbol{\pi}_\infty$

---

Based on Lemma 2 and the fact that $\pi^i \in \mathcal{U}_{\boldsymbol{\pi}}^i(\pi^i), \forall i \in \mathcal{N}, \pi^i \in \Pi^i$, we can know any HAML algorithm (weakly) improves the joint return at every iteration. In practical settings, such as deep MARL, the maximisation step of a HAML method can be performed by a few steps of gradient ascent on a sample average of HAMO (see Definition 1). We also highlight that if the neighbourhood operators $\mathcal{U}^i$ can be chosen so that they produce small policy-space subsets, then the resulting updates will be not only improving but also small. This, again, is a desirable property while optimising neural-network policies, as it helps stabilise the algorithm. One may wonder why the order of agents in HAML updates is randomised at every iteration; this condition has been necessary to establish convergence to NE, which is intuitively comprehendable: fixed-point joint policies of this randomised procedure assure that none of the agents is incentivised to make an update, namely reaching a NE. We provide the full list of the most fundamental HAML properties in Theorem 1 which shows that any method derived from Algorithm Template 1 solves the cooperative MARL problem.

**Theorem 1** (The Fundamental Theorem of Heterogeneous-Agent Mirror Learning). *Let, for every agent $i \in \mathcal{N}$, $\mathfrak{D}^{i,\nu}$ be a HADF, $\mathcal{U}^i$ be a neighbourhood operator, and let the sampling distributions $\beta_{\boldsymbol{\pi}}$ depend continuously on $\boldsymbol{\pi}$. Let $\boldsymbol{\pi}_0 \in \boldsymbol{\Pi}$, and the sequence of joint policies $(\boldsymbol{\pi}_k)_{k=0}^\infty$ be obtained by a HAML algorithm induced by $\mathfrak{D}^{i,\nu}, \mathcal{U}^i, \forall i \in \mathcal{N}$, and $\beta_{\boldsymbol{\pi}}$. Then, the joint policies induced by the algorithm enjoy the following list of properties*

    *1. Attain the monotonic improvement property,*

$$J(\boldsymbol{\pi}_{k+1}) \geq J(\boldsymbol{\pi}_k),$$

    *2. Their value functions converge to a Nash value function $V^{NE}$*

$$\lim_{k\to\infty} V_{\boldsymbol{\pi}_k} = V^{NE},$$

3. *Their expected returns converge to a Nash return,*

$$\lim_{k \to \infty} J(\boldsymbol{\pi}_k) = J^{NE},$$

4. *Their $\omega$-limit set consists of Nash equilibria.*

See the proof in Appendix C. With the above theorem, we can conclude that HAML provides a template for generating theoretically sound, stable, monotonically improving algorithms that enable agents to learn solving multi-agent cooperation tasks.

### 4.3 Existing HAML Instances: HATRPO and HAPPO

As a sanity check for its practicality, we show that two SOTA MARL methods—HATRPO and HAPPO (Kuba et al., 2022a)—are valid instances of HAML, which also provides an explanation for their excellent empirical performance.

We begin with an intuitive example of HATRPO, where agent $i_m$ (the permutation $i_{1:n}$ is drawn from the uniform distribution) updates its policy so as to maximise (in $\bar{\pi}^{i_m}$)

$$\mathbb{E}_{s \sim \rho_{\boldsymbol{\pi}_{\text{old}}}, \mathbf{a}^{i_{1:m-1}} \sim \boldsymbol{\pi}_{\text{new}}^{i_{1:m-1}}, a^{i_m} \sim \bar{\pi}^{i_m}} \left[ A_{\boldsymbol{\pi}_{\text{old}}}^{i_m}(s, \mathbf{a}^{i_{1:m-1}}, a^{i_m}) \right], \quad \text{subject to } \overline{D}_{\text{KL}}(\pi_{\text{old}}^{i_m}, \bar{\pi}^{i_m}) \le \delta.$$

This optimisation objective can be casted as a HAMO with the trivial HADF $\mathfrak{D}^{i_m, \nu} \equiv 0$, and the KL-divergence neighbourhood operator

$$\mathcal{U}_{\boldsymbol{\pi}}^{i_m}(\pi^{i_m}) = \left\{ \bar{\pi}^{i_m} \; \middle| \; \mathbb{E}_{s \sim \rho_{\boldsymbol{\pi}}} \left[ \text{KL}\left( \pi^{i_m}(\cdot^{i_m}|s), \bar{\pi}^{i_m}(\cdot^{i_m}|s) \right) \right] \le \delta \right\}.$$

The sampling distribution used in HATRPO is $\beta_{\boldsymbol{\pi}} = \rho_{\boldsymbol{\pi}}$. Lastly, as the agents update their policies in a random loop, the algorithm is an instance of HAML. Hence, it is monotonically improving and converges to a Nash equilibrium set.

In HAPPO, the update rule of agent $i_m$ is changes with respect to HATRPO as

$$\mathbb{E}_{s \sim \rho_{\boldsymbol{\pi}_{\text{old}}}, \mathbf{a}^{i_{1:m-1}} \sim \boldsymbol{\pi}_{\text{new}}^{i_{1:m-1}}, a^{i_m} \sim \pi_{\text{old}}^{i_m}} \left[ \min \left( \text{r}(\bar{\pi}^{i_m}) A_{\boldsymbol{\pi}_{\text{old}}}^{i_{1:m}}(s, \mathbf{a}^{i_{1:m}}), \text{clip}\left( \text{r}(\bar{\pi}^{i_m}), 1 \pm \epsilon \right) A_{\boldsymbol{\pi}_{\text{old}}}^{i_{1:m}}(s, \mathbf{a}^{i_{1:m}}) \right) \right],$$

where $\text{r}(\bar{\pi}^i) = \frac{\bar{\pi}^i(a^i|s)}{\pi_{\text{old}}^i(a^i|s)}$. We show in Appendix D that this optimisation objective is equivalent to

$$\mathbb{E}_{s \sim \rho_{\boldsymbol{\pi}_{\text{old}}}} \left[ \mathbb{E}_{\mathbf{a}^{i_{1:m-1}} \sim \boldsymbol{\pi}_{\text{new}}^{i_{1:m-1}}, a^{i_m} \sim \bar{\pi}^{i_m}} \left[ A_{\boldsymbol{\pi}_{\text{old}}}^{i_m}(s, \mathbf{a}^{i_{1:m-1}}, a^{i_m}) \right] \right.$$
$$\left. - \mathbb{E}_{\mathbf{a}^{i_{1:m}} \sim \pi_{\text{old}}^{i_{1:m}}} \left[ \text{ReLU}\left( \left[ \text{r}(\bar{\pi}^{i_m}) - \text{clip}\left( \text{r}(\bar{\pi}^{i_m}), 1 \pm \epsilon \right) \right] A_{\boldsymbol{\pi}_{\text{old}}}^{i_{1:m}}(s, \mathbf{a}^{i_{1:m}}) \right) \right] \right].$$

The purple term is clearly non-negative due to the presence of the ReLU funciton. Furthermore, for policies $\bar{\pi}^{i_m}$ sufficiently close to $\pi_{\text{old}}^{i_m}$, the clip operator does not activate, thus rendering $\text{r}(\bar{\pi}^{i_m})$ unchanged. Therefore, the purple term is zero at and in a region around $\bar{\pi}^{i_m} = \pi_{\text{old}}^{i_m}$, which also implies that its Gâteaux derivatives are zero. Hence, it evaluates a HADF for agent $i_m$, thus making HAPPO a valid HAML instance.

Finally, we would like to highlight that these results about HATRPO and HAPPO significantly strengthen the work originally by Kuba et al. (2022a) who only derived these learning protocols as approximations to a theoretically sound algorithm (see Algorithm 1 in Kuba et al. (2022a)), yet without such level of insights.

### 4.4 New HAML Instances: HAA2C and HADDPG

In this subsection, we exemplify how HAML can be used for derivation of principled MARL algorithms, and introduce heterogeneous-agent extensions of A2C and DDPG, different from those in Subsection 3.2. Our goal is not to refresh new SOTA performance on challenging tasks, but rather to verify the correctness of our theory, as well as deliver more robust versions of multi-agent extensions of popular RL algorithms such as A2C and DDPG.

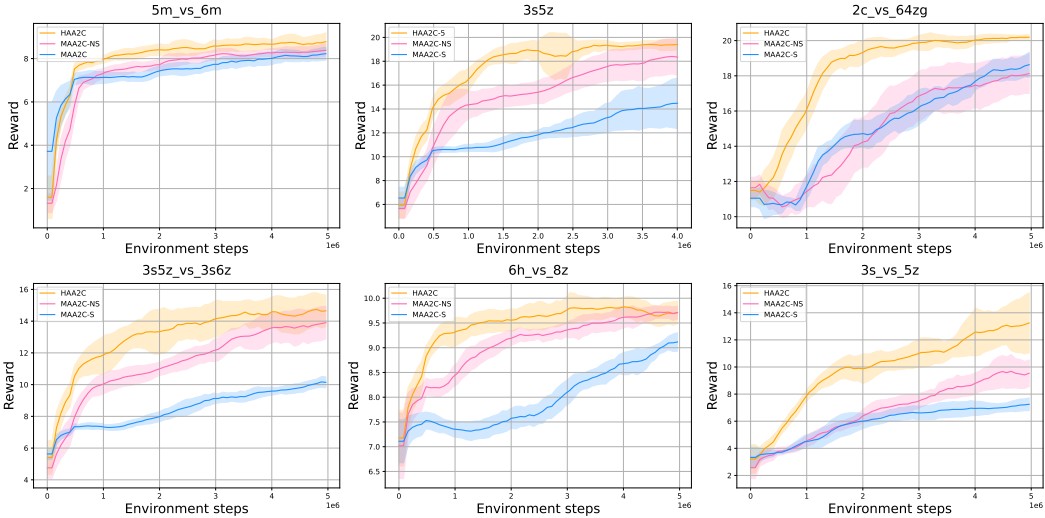

Figure 1: Comparison between HAA2C (yellow) vs MAA2C-S (blue) and MAA2C-NS (pink) in SMAC.

**HAA2C** intends to optimise the policy for the joint advantage function at every iteration, and similar to A2C, does not impose any penalties or constraints on that procedure. This learning procedure is accomplished by, first, drawing a random permutation of agents $i_{1:n}$, and then performing a few steps of gradient ascent on the objective of

$$\mathbb{E}_{s\sim\rho_{\boldsymbol{\pi}_{\text{old}}},\mathbf{a}^{i_{1:m}}\sim\boldsymbol{\pi}_{\text{old}}^{i_{1:m}}}\left[\frac{\boldsymbol{\pi}_{\text{new}}^{i_{1:m-1}}(\mathbf{a}^{i_{1:m-1}}|s)\pi^{i_m}(\mathbf{a}^{i_m}|s)}{\boldsymbol{\pi}_{\text{old}}^{i_{1:m-1}}(\mathbf{a}^{i_{1:m-1}}|s)\pi_{\text{old}}^{i_m}(\mathbf{a}^{i_m}|s)}A_{\boldsymbol{\pi}_{\text{old}}}^{i_m}(s,\mathbf{a}^{i_{1:m-1}},\mathbf{a}^{i_m})\right], \quad (6)$$

with respect to $\pi^{i_m}$ parameters, for each agent $i_m$ in the permutation, sequentially. In practice, we replace the multi-agent advantage $A_{\boldsymbol{\pi}_{\text{old}}}^{i_m}(s,\mathbf{a}^{i_{1:m-1}},\mathbf{a}^{i_m})$ with the joint advantage estimate which, thanks to the joint importance sampling in Equation (6), poses the same objective on the agent (see Appendix E for full pseudocode).

**HADDPG** aims to maximise the state-action value function off-policy. As it is a deterministic-action method, and thus importance sampling in its case translates to replacement of the old action inputs to the critic with the new ones. Namely, agent $i_m$ in a random permutation $i_{1:n}$ maximises

$$\mathbb{E}_{s\sim\beta_{\boldsymbol{\mu}_{\text{old}}}}\left[Q_{\boldsymbol{\mu}_{\text{old}}}^{i_{1:m}}\left(s,\boldsymbol{\mu}_{\text{new}}^{i_{1:m-1}}(s),\mu^{i_m}(s)\right)\right], \quad (7)$$

with respect to $\mu^{i_m}$, also with a few steps of gradient ascent. Similar to HAA2C, optimising the state-action value function (with the old action replacement) is equivalent to the original multi-agent value (see Appendix E for full pseudocode).

We are fully aware that none of these two methods exploits the entire abundance of the HAML framework—they do not possess HADFs or neighbourhood operators, as oppose to HATRPO or HAPPO. Thus, we speculate that the range of opportunities for HAML algorithm design is yet to be discovered with more future work. Nevertheless, we begin this search from these two straightforward methods, and analyse their performance in the next section.

## 5 EXPERIMENTS AND RESULTS

In this section[1], we challenge the capabilities of HAML in practice by testing HAA2C and HADDPG on the most challenging MARL benchmarks—we use StarCraft Multi-Agent Challenge (Samvelyan et al., 2019) and Multi-Agent MuJoCo (Peng et al., 2020), for discrete (only HAA2C) and continuous action settings, respectively.

We begin by demonstrating the performance of HAA2C. As a baseline, we use its "naive" predecessor, MAA2C, in both policy-sharing (MAA2C-S) and heterogeneous (MAA2C-NS) versions. Results on Figures 1 & 2 show that HAA2C generally achieves higher rewards in SMAC than both versions of MAA2C, while maintaining lower variance. The performance gap increases in MAMuJoCo, where the

---

[1]Our code is available at https://github.com/anonymouswater/HAML

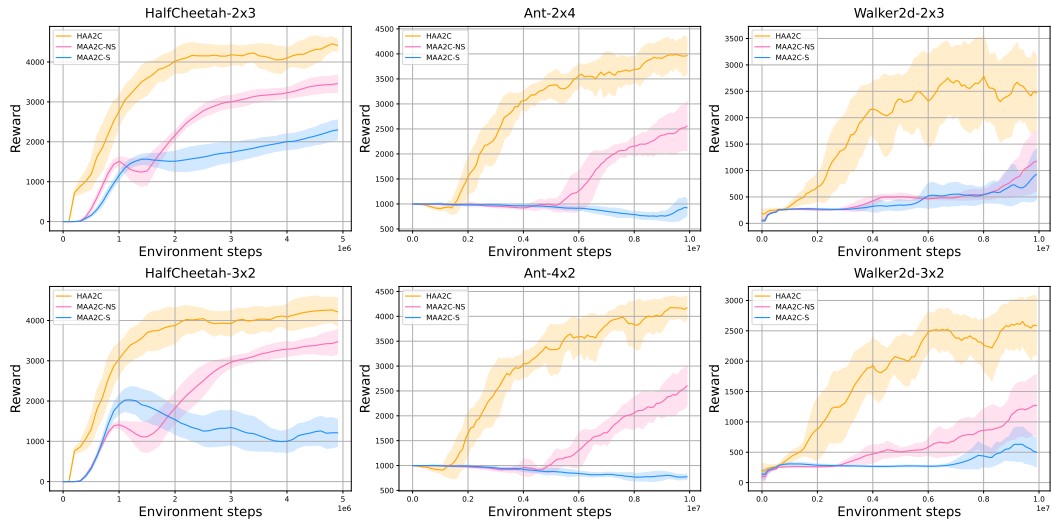

Figure 2: Comparison between HAA2C (yellow) vs MAA2C-S (blue) and MAA2C-NS (pink) in MAMuJoCo.

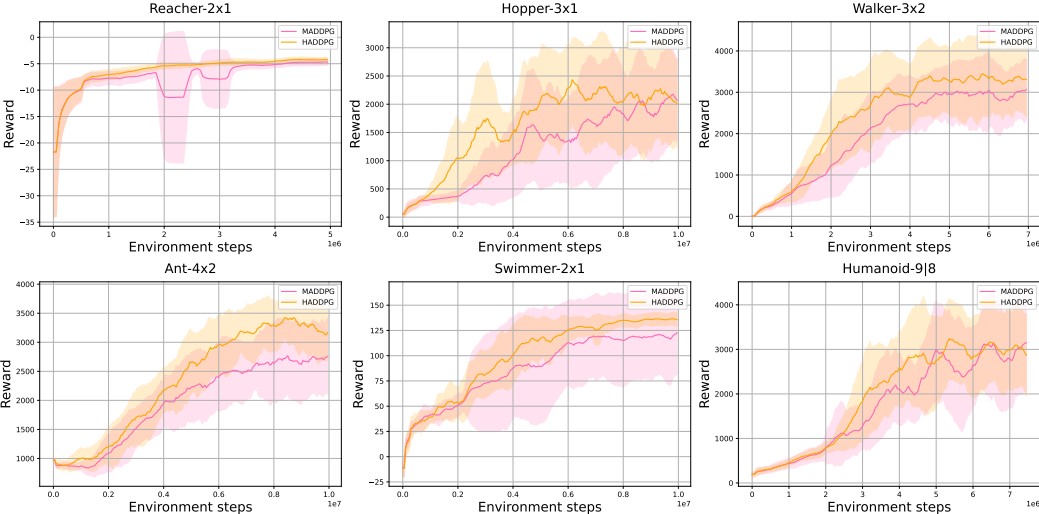

Figure 3: Comparison between HADDPG (yellow) and MADDPG (pink) in MAMuJoCo.

agents must learn diverse policies to master complex movements (Kuba et al., 2022a). Here, echoing Proposition 1, the homogeneous MAA2C-S fails completely, and conventional, heterogeneous MAA2C-NS underperforms by a significant margin (recall Proposition 2).

As HADDPG is a continuous-action method, we test it only on MAMuJoCo (precisely 6 tasks) and compare it to MADDPG. Figure 3 revelas that HADDPG achieves higher reward than MADDPG, while sometimes displaying significantly lower variance (*e.g* in Reacher-2x1 and Swimmer-2x1). Hence, we conclude that HADDPG performs better.

## 6 CONCLUSION

In this paper, we described and addressed the problem of lacking principled treatments for cooperative MARL tasks. Our main contribution is the development of heterogeneous-agent mirror learning (HAML), a class of provably correct MARL algorithms, whose properties are rigorously profiled. We verified the correctness and the practicality of HAML by interpreting current SOTA methods—HATRPO and HAPPO—as HAML instances and also by deriving and testing heterogeneous-agent extensions of successful RL algorithms, named as HAA2C and HADDPG. We expect HAML to help create a template for designing both principled and practical MARL algorithms hereafter.

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

# A    PROOFS OF PRELIMINARY RESULTS

## A.1    PROOF OF LEMMA 1

**Lemma 1** (Multi-Agent Advantage Decomposition). *Let $\boldsymbol{\pi}$ be a joint policy, and $i_1, \ldots, i_m$ be an arbitrary ordered subset of agents. Then, for any state $s$ and joint action $\boldsymbol{a}^{i_{1:m}}$,*

$$A_{\boldsymbol{\pi}}^{i_{1:m}}\left(s, \boldsymbol{a}^{i_{1:m}}\right) = \sum_{j=1}^{m} A_{\boldsymbol{\pi}}^{i_j}\left(s, \boldsymbol{a}^{i_{1:j-1}}, a^{i_j}\right). \tag{1}$$

*Proof.* (We quote the proof from (Kuba et al., 2022a).) We start as expressing the multi-agent advantage as a telescoping sum, and then rewrite it using the definition of multi-agent advantage,

$$\begin{aligned}
A_{\boldsymbol{\pi}}^{i_{1:m}}(s, \boldsymbol{a}^{i_{1:m}}) &= Q_{\boldsymbol{\pi}}^{i_{1:m}}(s, \boldsymbol{a}^{i_{1:m}}) - V_{\boldsymbol{\pi}}(s) \\
&= \sum_{j=1}^{m} \left[ Q_{\boldsymbol{\pi}}^{i_{1:j}}(s, \boldsymbol{a}^{i_{1:j}}) - Q_{\boldsymbol{\pi}}^{i_{1:j-1}}(s, \boldsymbol{a}^{i_{1:j-1}}) \right] = \sum_{j=1}^{m} A_{\boldsymbol{\pi}}^{i_j}(s, \boldsymbol{a}^{i_{1:j-1}}, a^{i_j}).
\end{aligned}$$

$\square$

## A.2    PROOF OF PROPOSITION 2

**Proposition 2** (Trap of Heterogeneity). *Let's consider a fully-cooperative game with $2$ agents, one state, and the joint action space $\{0,1\}^2$, where the reward is given by $r(0,0) = 0, r(0,1) = r(1,0) = 2$, and $r(1,1) = -1$. Suppose that $\pi_{old}^i(0) > 0.6$ for $i = 1, 2$. Then, if agents $i$ update their policies by*

$$\pi_{new}^i = \arg\max_{\pi^i} \mathbb{E}_{a^i \sim \pi^i, a^{-i} \sim \pi_{old}^{-i}} \left[ A_{\boldsymbol{\pi}_{old}}(a^i, a^{-i}) \right], \forall i \in \mathcal{N},$$

*then the resulting policy will yield a lower return,*

$$J(\boldsymbol{\pi}_{old}) > J(\boldsymbol{\pi}_{new}) = \min_{\boldsymbol{\pi}} J(\boldsymbol{\pi}).$$

As there is only one state, we can ignore the infinite horizon and the discount factor $\gamma$, thus making the state-action value and the reward functions equivalent, $Q \equiv r$.

Let us, for brevity, define $\pi^i = \pi_{old}^i(0) > 0.6$, for $i = 1, 2$. We have

$$\begin{aligned}
J(\boldsymbol{\pi}_{old}) &= \Pr(a^1 = a^2 = 0)r(0,0) + \left(1 - \Pr(a^1 = a^2 = 0)\right)\mathbb{E}[r(a^1, a^2)|(a^1, a^2) \neq (0,0)] \\
&> 0.6^2 \times 0 - (1 - 0.6^2) = -0.64.
\end{aligned}$$

The update rule stated in the proposition can be equivalently written as

$$\pi_{new}^i = \arg\max_{\pi^i} \mathbb{E}_{a^i \sim \pi^i, a^{-i} \sim \pi_{old}^{-i}} \left[ Q_{\boldsymbol{\pi}_{old}}(a^i, a^{-i}) \right]. \tag{8}$$

We have

$$\mathbb{E}_{a^{-i} \sim \pi_{old}^{-i}} \left[ Q_{\boldsymbol{\pi}_{old}}(0, a^{-i}) \right] = \pi^{-i}Q(0,0) + (1 - \pi^{-i})Q(0,1) = \pi^{-i}r(0,0) + (1 - \pi^{-i})r(0,1) = 2(1 - \pi^{-i}),$$

and similarly

$$\mathbb{E}_{a^{-i} \sim \pi_{old}^{-i}} \left[ Q_{\boldsymbol{\pi}_{old}}(1, a^{-i}) \right] = \pi^{-i}r(1,0) + (1 - \pi^{-i})r(1,1) = 2\pi^{-i} - (1 - \pi^{-i}) = 3\pi^{-i} - 1.$$

Hence, if $\pi^{-i} > 0.6$, then

$$\mathbb{E}_{a^{-i} \sim \pi_{old}^{-i}} \left[ Q_{\boldsymbol{\pi}_{old}}(1, a^{-i}) \right] = 3\pi^{-i} - 1 > 3 \times 0.6 - 1 = 0.8 > 2 - 2\pi^{-i} = \mathbb{E}_{a^{-i} \sim \pi_{old}^{-i}} \left[ Q_{\boldsymbol{\pi}_{old}}(0, a^{-i}) \right].$$

Therefore, for every $i$, the solution to Equation (8) is the greedy policy $\pi_{new}^i(1) = 1$. Therefore,

$$J(\boldsymbol{\pi}_{new}) = Q(1,1) = r(1,1) = -1,$$

which finishes the proof.

# B  PROOF OF *HAMO Is All You Need* LEMMA

**Lemma 2** (HAMO Is All You Need). *Let $\boldsymbol{\pi}_{old}$ and $\boldsymbol{\pi}_{new}$ be joint policies and let $i_{1:n} \in \mathrm{Sym}(n)$ be an agent permutation. Suppose that, for every state $s \in \mathcal{S}$ and every $m = 1, \ldots, n$,*

$$\big[\mathcal{M}^{(\pi_{new}^{i_m})}_{\mathfrak{D}^{i_m},\nu,\boldsymbol{\pi}_{new}^{i_{1:m-1}}} A_{\boldsymbol{\pi}_{old}}\big](s) \geq \big[\mathcal{M}^{(\pi_{old}^{i_m})}_{\mathfrak{D}^{i_m},\nu,\boldsymbol{\pi}_{new}^{i_{1:m-1}}} A_{\boldsymbol{\pi}_{old}}\big](s). \tag{5}$$

*Then, $\boldsymbol{\pi}_{new}$ is jointly better than $\boldsymbol{\pi}_{old}$, so that for every state $s$,*

$$V_{\boldsymbol{\pi}_{new}}(s) \geq V_{\boldsymbol{\pi}_{old}}(s).$$

*Proof.* Let $\widetilde{\mathfrak{D}}_{\boldsymbol{\pi}_{\text{old}}}(\boldsymbol{\pi}_{\text{new}}|s) \triangleq \sum_{m=1}^{n} \frac{\nu^{i_m}_{\boldsymbol{\pi}_{\text{old}},\hat{\pi}^{i_m}}(s)}{\beta_{\boldsymbol{\pi}_{\text{old}}}(s)} \mathfrak{D}^{i_m}_{\boldsymbol{\pi}_{\text{old}}}(\pi^{i_m}_{\text{new}}|s, \boldsymbol{\pi}^{i_{1:m-1}}_{\text{new}})$. Combining this with Lemma 1 gives

$$\mathbb{E}_{\mathbf{a} \sim \boldsymbol{\pi}_{\text{new}}}\big[A_{\boldsymbol{\pi}_{\text{old}}}(s, \mathbf{a})\big] - \widetilde{\mathfrak{D}}_{\boldsymbol{\pi}_{\text{old}}}(\boldsymbol{\pi}_{\text{new}}|s)$$

$$= \sum_{m=1}^{n} \mathbb{E}_{\mathbf{a}^{i_{1:m-1}} \sim \boldsymbol{\pi}^{i_{1:m-1}}_{\text{new}}, \mathbf{a}^{i_m} \sim \pi^{i_m}_{\text{new}}} \Big[A^{i_m}_{\boldsymbol{\pi}_{\text{old}}}(s, \boldsymbol{a}^{i_{1:m-1}}, a^{i_m}) - \frac{\nu^{i_m}_{\boldsymbol{\pi}_{\text{old}},\pi^{i_m}_{\text{new}}}(s)}{\beta_{\boldsymbol{\pi}_{\text{old}}}(s)} \mathfrak{D}^{i_m}_{\boldsymbol{\pi}_{\text{old}}}(\pi^{i_m}_{\text{new}}|s, \boldsymbol{\pi}^{i_{1:m-1}}_{\text{new}})\Big]$$

by Inequality (5)

$$\geq \sum_{m=1}^{n} \mathbb{E}_{\mathbf{a}^{i_{1:m-1}} \sim \boldsymbol{\pi}^{i_{1:m-1}}_{\text{new}}, \mathbf{a}^{i_m} \sim \pi^{i_m}_{\text{old}}} \Big[A^{i_m}_{\boldsymbol{\pi}_{\text{old}}}(s, \boldsymbol{a}^{i_{1:m-1}}, a^{i_m}) - \frac{\nu^{i_m}_{\boldsymbol{\pi}_{\text{old}},\pi^{i_m}_{\text{old}}}(s)}{\beta_{\boldsymbol{\pi}_{\text{old}}}(s)} \mathfrak{D}^{i_m}_{\boldsymbol{\pi}_{\text{old}}}(\pi^{i_m}_{\text{old}}|s, \boldsymbol{\pi}^{i_{1:m-1}}_{\text{new}})\Big]$$

$$= \mathbb{E}_{\mathbf{a} \sim \boldsymbol{\pi}_{\text{old}}}\big[A_{\boldsymbol{\pi}_{\text{old}}}(s, \mathbf{a})\big] - \widetilde{\mathfrak{D}}_{\boldsymbol{\pi}_{\text{old}}}(\boldsymbol{\pi}_{\text{old}}|s).$$

The resulting inequality can be equivalently rewritten as

$$\mathbb{E}_{\mathbf{a} \sim \boldsymbol{\pi}_{\text{new}}}\big[Q_{\boldsymbol{\pi}_{\text{old}}}(s, \mathbf{a})\big] - \widetilde{\mathfrak{D}}_{\boldsymbol{\pi}_{\text{old}}}(\boldsymbol{\pi}_{\text{new}}|s) \geq \mathbb{E}_{\mathbf{a} \sim \boldsymbol{\pi}_{\text{old}}}\big[Q_{\boldsymbol{\pi}_{\text{old}}}(s, \mathbf{a})\big] - \widetilde{\mathfrak{D}}_{\boldsymbol{\pi}_{\text{old}}}(\boldsymbol{\pi}_{\text{old}}|s), \forall s \in \mathcal{S}. \tag{9}$$

We use it to prove the claim as follows,

$$V_{\boldsymbol{\pi}_{\text{new}}}(s) = \mathbb{E}_{\mathbf{a} \sim \boldsymbol{\pi}_{\text{new}}}\big[Q_{\boldsymbol{\pi}_{\text{new}}}(s, \mathbf{a})\big]$$

$$= \mathbb{E}_{\mathbf{a} \sim \boldsymbol{\pi}_{\text{new}}}\big[Q_{\boldsymbol{\pi}_{\text{old}}}(s, \mathbf{a})\big] - \widetilde{\mathfrak{D}}_{\boldsymbol{\pi}_{\text{old}}}(\boldsymbol{\pi}_{\text{new}}|s)$$

$$\quad + \widetilde{\mathfrak{D}}_{\boldsymbol{\pi}_{\text{old}}}(\boldsymbol{\pi}_{\text{new}}|s) + \mathbb{E}_{\mathbf{a} \sim \boldsymbol{\pi}_{\text{new}}}\big[Q_{\boldsymbol{\pi}_{\text{new}}}(s, \mathbf{a}) - Q_{\boldsymbol{\pi}_{\text{old}}}(s, \mathbf{a})\big],$$

by Inequality (9)

$$\geq \mathbb{E}_{\mathbf{a} \sim \boldsymbol{\pi}_{\text{old}}}\big[Q_{\boldsymbol{\pi}_{\text{old}}}(s, \mathbf{a})\big] - \widetilde{\mathfrak{D}}_{\boldsymbol{\pi}_{\text{old}}}(\boldsymbol{\pi}_{\text{old}}|s)$$

$$\quad + \widetilde{\mathfrak{D}}_{\boldsymbol{\pi}_{\text{old}}}(\boldsymbol{\pi}_{\text{new}}|s) + \mathbb{E}_{\mathbf{a} \sim \boldsymbol{\pi}_{\text{new}}}\big[Q_{\boldsymbol{\pi}_{\text{new}}}(s, \mathbf{a}) - Q_{\boldsymbol{\pi}_{\text{old}}}(s, \mathbf{a})\big],$$

$$= V_{\boldsymbol{\pi}_{\text{old}}}(s) + \widetilde{\mathfrak{D}}_{\boldsymbol{\pi}_{\text{old}}}(\boldsymbol{\pi}_{\text{new}}|s) + \mathbb{E}_{\mathbf{a} \sim \boldsymbol{\pi}_{\text{new}}}\big[Q_{\boldsymbol{\pi}_{\text{new}}}(s, \mathbf{a}) - Q_{\boldsymbol{\pi}_{\text{old}}}(s, \mathbf{a})\big]$$

$$= V_{\boldsymbol{\pi}_{\text{old}}}(s) + \widetilde{\mathfrak{D}}_{\boldsymbol{\pi}_{\text{old}}}(\boldsymbol{\pi}_{\text{new}}|s) + \mathbb{E}_{\mathbf{a} \sim \boldsymbol{\pi}_{\text{new}}, s' \sim P}\big[r(s, \mathbf{a}) + \gamma V_{\boldsymbol{\pi}_{\text{new}}}(s') - r(s, \mathbf{a}) - \gamma V_{\boldsymbol{\pi}_{\text{old}}}(s')\big]$$

$$= V_{\boldsymbol{\pi}_{\text{old}}}(s) + \widetilde{\mathfrak{D}}_{\boldsymbol{\pi}_{\text{old}}}(\boldsymbol{\pi}_{\text{new}}|s) + \gamma \mathbb{E}_{\mathbf{a} \sim \boldsymbol{\pi}_{\text{new}}, s' \sim P}\big[V_{\boldsymbol{\pi}_{\text{new}}}(s') - V_{\boldsymbol{\pi}_{\text{old}}}(s')\big]$$

$$\geq V_{\boldsymbol{\pi}_{\text{old}}}(s) + \gamma \inf_{s'}\big[V_{\boldsymbol{\pi}_{\text{new}}}(s') - V_{\boldsymbol{\pi}_{\text{old}}}(s')\big].$$

Hence  $V_{\boldsymbol{\pi}_{\text{new}}}(s) - V_{\boldsymbol{\pi}_{\text{old}}}(s) \geq \gamma \inf_{s'}\big[V_{\boldsymbol{\pi}_{\text{new}}}(s') - V_{\boldsymbol{\pi}_{\text{old}}}(s')\big].$

Taking infimum over $s$ and simplifying

$$(1 - \gamma) \inf_{s}\big[V_{\boldsymbol{\pi}_{\text{new}}}(s) - V_{\boldsymbol{\pi}_{\text{old}}}(s)\big] \geq 0.$$

Therefore, $\inf_{s}\big[V_{\boldsymbol{\pi}_{\text{new}}}(s) - V_{\boldsymbol{\pi}_{\text{old}}}(s)\big] \geq 0$, which proves the lemma.  □

## C  PROOF OF THEOREM 1

**Lemma 3.** *Suppose an agent $i_m$ maximises the expected HAMO*

$$\pi_{new}^{i_m} = \arg\max_{\pi^{i_m} \in \mathcal{U}_{\boldsymbol{\pi}_{old}}^{i_m}(\pi_{old}^{i_m})} \mathbb{E}_{\mathbf{s} \sim \beta_{\boldsymbol{\pi}_{old}}}\left[\left[\mathcal{M}_{\mathfrak{D}^{i_m},\nu,\boldsymbol{\pi}_{new}^{i_{1:m-1}}}^{(\pi^{i_m})} A_{\boldsymbol{\pi}_{old}}\right](\mathbf{s})\right]. \tag{10}$$

*Then, for every state $s \in \mathcal{S}$*

$$\left[\mathcal{M}_{\mathfrak{D}^{i_m},\nu,\boldsymbol{\pi}_{new}^{i_{1:m-1}}}^{(\pi_{new}^{i_m})} A_{\boldsymbol{\pi}_{old}}\right](s) \geq \left[\mathcal{M}_{\mathfrak{D}^{i_m},\nu,\boldsymbol{\pi}_{new}^{i_{1:m-1}}}^{(\pi_{old}^{i_m})} A_{\boldsymbol{\pi}_{old}}\right](s).$$

*Proof.* We will prove this statement by contradiction. Suppose that there exists $s_0 \in \mathcal{S}$ such that

$$\left[\mathcal{M}_{\mathfrak{D}^{i_m},\nu,\boldsymbol{\pi}_{\text{new}}^{i_{1:m-1}}}^{(\pi_{\text{new}}^{i_m})} A_{\boldsymbol{\pi}_{\text{old}}}\right](s_0) < \left[\mathcal{M}_{\mathfrak{D}^{i_m},\nu,\boldsymbol{\pi}_{\text{new}}^{i_{1:m-1}}}^{(\pi_{\text{old}}^{i_m})} A_{\boldsymbol{\pi}_{\text{old}}}\right](s_0). \tag{11}$$

Let us define the following policy $\hat{\pi}^{i_m}$.

$$\hat{\pi}^{i_m}(\cdot^{i_m}|s) = \begin{cases} \pi_{\text{old}}^{i_m}(\cdot^{i_m}|s), & \text{at } s = s_0 \\ \pi_{\text{new}}^{i_m}(\cdot^{i_m}|s), & \text{at } s \neq s_0 \end{cases}$$

Note that $\hat{\pi}^{i_m}$ is (weakly) closer to $\pi_{\text{old}}^{i_m}$ than $\pi_{\text{new}}^{i_m}$ at $s_0$, and at the same distance at other states. Together with $\pi_{\text{new}}^{i_m} \in \mathcal{U}_{\boldsymbol{\pi}_{\text{old}}}^{i_m}(\pi_{\text{old}}^{i_m})$, this implies that $\hat{\pi}^{i_m} \in \mathcal{U}_{\boldsymbol{\pi}_{\text{old}}}^{i_m}(\pi_{\text{old}}^{i_m})$. Further,

$$\mathbb{E}_{\mathbf{s}\sim\beta_{\boldsymbol{\pi}_{\text{old}}}}\left[\left[\mathcal{M}_{\mathfrak{D}^{i_m},\nu,\boldsymbol{\pi}_{\text{new}}^{i_{1:m-1}}}^{(\hat{\pi}^{i_m})} A_{\boldsymbol{\pi}_{\text{old}}}\right](\mathbf{s})\right] - \mathbb{E}_{\mathbf{s}\sim\beta_{\boldsymbol{\pi}_{\text{old}}}}\left[\left[\mathcal{M}_{\mathfrak{D}^{i_m},\nu,\boldsymbol{\pi}_{\text{new}}^{i_{1:m-1}}}^{(\pi_{\text{new}}^{i_m})} A_{\boldsymbol{\pi}_{\text{old}}}\right](\mathbf{s})\right]$$

$$\beta_{\boldsymbol{\pi}_{\text{old}}}(s_0)\left(\left[\mathcal{M}_{\mathfrak{D}^{i_m},\nu,\boldsymbol{\pi}_{\text{new}}^{i_{1:m-1}}}^{(\hat{\pi}^{i_m})} A_{\boldsymbol{\pi}_{\text{old}}}\right](s_0) - \left[\mathcal{M}_{\mathfrak{D}^{i_m},\nu,\boldsymbol{\pi}_{\text{new}}^{i_{1:m-1}}}^{(\hat{\pi}^{i_m})} A_{\boldsymbol{\pi}_{\text{old}}}\right](s_0)\right) > 0.$$

The above contradicts $\pi_{\text{new}}^{i_m}$ as being the argmax of Inequality (11), as $\hat{\pi}^{i_m}$ is strictly better. The contradiciton finishes the proof. □

**Theorem 1** (The Fundamental Theorem of Heterogeneous-Agent Mirror Learning). *Let, for every agent $i \in \mathcal{N}$, $\mathfrak{D}^{i,\nu}$ be a HADF, $\mathcal{U}^i$ be a neighbourhood operator, and let the sampling distributions $\beta_{\boldsymbol{\pi}}$ depend continuously on $\boldsymbol{\pi}$. Let $\boldsymbol{\pi}_0 \in \Pi$, and the sequence of joint policies $(\boldsymbol{\pi}_k)_{k=0}^{\infty}$ be obtained by a HAML algorithm induced by $\mathfrak{D}^{i,\nu}, \mathcal{U}^i, \forall i \in \mathcal{N}$, and $\beta_{\boldsymbol{\pi}}$. Then, the joint policies induced by the algorithm enjoy the following list of properties*

1. *Attain the monotonic improvement property,*

$$J(\boldsymbol{\pi}_{k+1}) \geq J(\boldsymbol{\pi}_k),$$

2. *Their value functions converge to a Nash value function $V^{NE}$*

$$\lim_{k\to\infty} V_{\boldsymbol{\pi}_k} = V^{NE},$$

3. *Their expected returns converge to a Nash return,*

$$\lim_{k\to\infty} J(\boldsymbol{\pi}_k) = J^{NE},$$

4. *Their $\omega$-limit set consists of Nash equilibria.*

*Proof.* **Proof of Property 1.**
It follows from combining Lemmas 2 & 3.

**Proof of Properties 2, 3 & 4.**
**Step 1: convergence of the value function.** By Lemma 2, we have that $V_{\boldsymbol{\pi}_k}(s) \leq V_{\boldsymbol{\pi}_{k+1}}(s)$, $\forall s \in \mathcal{S}$, and that the value function is upper-bounded by $V_{\max}$. Hence, the sequence of value functions $(V_{\boldsymbol{\pi}_k})_{k\in\mathbb{N}}$ converges. We denote its limit by $V$.

**Step 2: characterisation of limit points.** As the joint policy space $\Pi$ is bounded, by Bolzano-Weierstrass theorem, we know that the sequence $(\boldsymbol{\pi}_k)_{k\in\mathbb{N}}$ has a convergent subsequence. Therefore,

it has at least one limit point policy. Let $\bar{\boldsymbol{\pi}}$ be such a limit point. We introduce an auxiliary notation: for a joint policy $\boldsymbol{\pi}$ and a permutation $i_{1:n}$, let $\mathrm{HU}(\boldsymbol{\pi}, i_{1:n})$ be a joint policy obtained by a HAML update from $\boldsymbol{\pi}$ along the permutation $i_{1:n}$.

**Claim:** For any permutation $z_{1:n} \in \mathrm{Sym}(n)$,

$$\bar{\boldsymbol{\pi}} = \mathrm{HU}(\bar{\boldsymbol{\pi}}, z_{1:n}). \tag{12}$$

**Proof of Claim.** Let $\hat{\boldsymbol{\pi}} = \mathrm{HU}(\bar{\boldsymbol{\pi}}, z_{1:n}) \neq \bar{\boldsymbol{\pi}}$ and $(\boldsymbol{\pi}_{k_r})_{r \in \mathbb{N}}$ be a subsequence converging to $\bar{\boldsymbol{\pi}}$. Let us recall that the limit value function is unique and denoted as $V$. Writing $\mathbb{E}_{i_{1:n}^{0:\infty}}[\cdot]$ for the expectation operator under the stochastic process $(i_{1:n}^k)_{k \in \mathbb{N}}$ of update orders, for a state $s \in \mathcal{S}$, we have

$$0 = \lim_{r \to \infty} \mathbb{E}_{i_{1:n}^{0:\infty}}\big[V_{\boldsymbol{\pi}_{k_r+1}}(s) - V_{\boldsymbol{\pi}_{k_r}}(s)\big]$$

as every choice of permutation improves the value function

$$\geq \lim_{r \to \infty} \mathrm{P}(i_{1:n}^{k_r} = z_{1:n})\big[V_{\mathrm{HU}(\boldsymbol{\pi}_{k_r}, z_{1:n})}(s) - V_{\boldsymbol{\pi}_{k_r}}(s)\big]$$

$$= p(z_{1:n}) \lim_{r \to \infty} \big[V_{\mathrm{HU}(\boldsymbol{\pi}_{k_r}, z_{1:n})}(s) - V_{\boldsymbol{\pi}_{k_r}}(s)\big].$$

By the continuity of the expected HAMO (following from the continuity of the value function (Kuba et al., 2022a, Appendix A), HADFs, neighbourhood operators, and the sampling distribution) we obtain that the first component of $\mathrm{HU}(\boldsymbol{\pi}_{k_r}, z_{1:n})$, which is $\pi_{k_r+1}^{z_1}$, is continuous in $\boldsymbol{\pi}_{k_r}$ by Berge's Maximum Theorem (Ausubel & Deneckere, 1993). Applying this argument recursively for $z_2, \ldots, z_n$, we have that $\mathrm{HU}(\boldsymbol{\pi}_{k_r}, z_{1:n})$ is continuous in $\boldsymbol{\pi}_{k_r}$. Hence, as $\boldsymbol{\pi}_{k_r}$ converges to $\bar{\boldsymbol{\pi}}$, its HU converges to the HU of $\bar{\boldsymbol{\pi}}$, which is $\hat{\boldsymbol{\pi}}$. Hence, we continue wiriting the above derivation as

$$= p(z_{1:n})\big[V_{\hat{\boldsymbol{\pi}}}(s) - V_{\bar{\boldsymbol{\pi}}}(s)\big] \geq 0, \text{ by Lemma 2.}$$

As $s$ was arbitrary, the state-value function of $\hat{\boldsymbol{\pi}}$ is the same as that of $\boldsymbol{\pi}$: $V_{\hat{\boldsymbol{\pi}}} = V_{\boldsymbol{\pi}}$, by the Bellman equation (Sutton & Barto, 2018): $Q(s, \boldsymbol{a}) = r(s, \boldsymbol{a}) + \gamma \mathbb{E} V(s')$, this also implies that their state-value and advantage functions are the same: $Q_{\hat{\boldsymbol{\pi}}} = Q_{\bar{\boldsymbol{\pi}}}$ and $A_{\hat{\boldsymbol{\pi}}} = A_{\bar{\boldsymbol{\pi}}}$. Let $m$ be the smallest integer such that $\hat{\pi}^{z_m} \neq \bar{\pi}^{z_m}$. This means that $\hat{\pi}^{z_m}$ achieves a greater expected HAMO than $\bar{\pi}^{z_m}$, for which it is zero. Hence,

$$0 < \mathbb{E}_{s \sim \beta_{\boldsymbol{\pi}}}\Big[\big[\mathcal{M}_{\mathfrak{D}^{z,\nu}, \bar{\pi}^{z_{1:m-1}}}^{(\hat{\pi}^{z_m})} A_{\bar{\boldsymbol{\pi}}}\big](s)\Big]$$

$$= \mathbb{E}_{s \sim \beta_{\boldsymbol{\pi}}}\Big[\mathbb{E}_{\mathbf{a}^{z_{1:m}} \sim \bar{\pi}^{z_{1:m-1}}, \mathbf{a}^{z_m} \sim \hat{\pi}^{z_m}}\big[A_{\bar{\boldsymbol{\pi}}}^{z_m}(s, \mathbf{a}^{z_{1:m-1}}, \mathbf{a}^{z_m})\big] - \frac{\nu_{\bar{\boldsymbol{\pi}}, \hat{\pi}^{z_m}}^{z_m}(s)}{\beta_{\bar{\boldsymbol{\pi}}}(s)} \mathfrak{D}_{\boldsymbol{\pi}}^{z_m}(\hat{\pi}^{z_m}|s, \bar{\boldsymbol{\pi}}^{z_{1:m-1}})\Big]$$

$$= \mathbb{E}_{s \sim \beta_{\boldsymbol{\pi}}}\Big[\mathbb{E}_{\mathbf{a}^{z_{1:m}} \sim \bar{\pi}^{z_{1:m-1}}, \mathbf{a}^{z_m} \sim \hat{\pi}^{z_m}}\big[A_{\hat{\boldsymbol{\pi}}}^{z_m}(s, \mathbf{a}^{z_{1:m-1}}, \mathbf{a}^{z_m})\big] - \frac{\nu_{\bar{\boldsymbol{\pi}}, \hat{\pi}^{z_m}}^{z_m}(s)}{\beta_{\bar{\boldsymbol{\pi}}}(s)} \mathfrak{D}_{\boldsymbol{\pi}}^{z_m}(\hat{\pi}^{z_m}|s, \bar{\boldsymbol{\pi}}^{z_{1:m-1}})\Big]$$

and as the expected value of the multi-agent advantage function is zero

$$= \mathbb{E}_{s \sim \beta_{\boldsymbol{\pi}}}\Big[- \frac{\nu_{\bar{\boldsymbol{\pi}}, \hat{\pi}^{z_m}}^{z_m}(s)}{\beta_{\bar{\boldsymbol{\pi}}}(s)} \mathfrak{D}_{\boldsymbol{\pi}}^{z_m}(\hat{\pi}^{z_m}|s, \bar{\boldsymbol{\pi}}^{z_{1:m-1}})\Big] \leq 0.$$

This is a contradiction, and so the claim in Equation (12) is proved, and the Step 2 is finished.

**Step 3: dropping the HADF.** Consider an arbitrary limit point joint policy $\bar{\boldsymbol{\pi}}$. By Step 2, for any permutation $i_{1:n}$, considering the first component of the HU, and writing $\nu^i = \nu_{\bar{\boldsymbol{\pi}}, \pi^i}^i$,

$$\bar{\pi}^{i_1} = \max_{\pi^{i_1} \in \mathcal{U}_{\boldsymbol{\pi}}^{i_1}(\pi^{i_1})} \mathbb{E}_{s \sim \beta_{\bar{\boldsymbol{\pi}}}}\Big[\big[\mathcal{M}_{\mathfrak{D}^{i_1,\nu}}^{(\pi^{i_1})} A_{\bar{\boldsymbol{\pi}}}\big](s)\Big] \tag{13}$$

$$= \max_{\pi^{i_1} \in \mathcal{U}_{\boldsymbol{\pi}}^{i_1}(\pi^{i_1})} \mathbb{E}_{s \sim \beta_{\bar{\boldsymbol{\pi}}}}\Big[\mathbb{E}_{\mathbf{a}^{i_1} \sim \pi^{i_1}}\big[A_{\bar{\boldsymbol{\pi}}}(s, \mathbf{a}^{i_1})\big] - \frac{\nu^{i_1}(s)}{\beta_{\bar{\boldsymbol{\pi}}}(s)} \mathfrak{D}_{\bar{\boldsymbol{\pi}}}^{i_1}(\pi^{i_1}|s)\Big].$$

As the HADF is non-negative, and at $\pi^{i_1} = \bar{\pi}^{i_1}$ its value and of its all Gâteaux derivatives are zero, it follows by Step 3 of Theorem 1 of (Kuba et al., 2022b) that for every $s \in \mathcal{S}$,

$$\bar{\pi}^{i_1}(\cdot^{i_m}|s) = \underset{\pi^{i_1} \in \mathcal{P}(\mathcal{A}^{i_1})}{\arg\max} \mathbb{E}_{\mathbf{a}^{i_1} \sim \pi^{i_1}}\big[Q_{\bar{\boldsymbol{\pi}}}^{i_1}(s, \mathbf{a}^{i_1})\big].$$

**Step 4: Nash equilibrium.** We have proved that $\bar{\pi}$ satisfies

$$
\begin{aligned}
\bar{\pi}^i(\cdot^i|s) &= \underset{\pi^i(\cdot^i|s)\in\mathcal{P}(\mathcal{A}^i)}{\arg\max} \mathbb{E}_{\mathbf{a}^i\sim\pi^i}\big[Q^i_{\bar{\pi}}(s,\mathbf{a}^i)\big] \\
&= \underset{\pi^i(\cdot^i|s)\in\mathcal{P}(\mathcal{A}^i)}{\arg\max} \mathbb{E}_{\mathbf{a}^i\sim\pi^i,\mathbf{a}^{-i}\sim\bar{\pi}^{-i}}\big[Q_{\bar{\pi}}(s,\mathbf{a})\big], \; \forall i\in\mathcal{N}, s\in\mathcal{S}.
\end{aligned}
$$

Hence, by considering $\bar{\pi}^{-i}$ fixed, we see that $\bar{\pi}^i$ satisfies the condition for the optimal policy (Sutton & Barto, 2018), and hence

$$
\bar{\pi}^i = \underset{\pi^i\in\Pi^i}{\arg\max} \, J(\pi^i,\bar{\pi}^{-i}).
$$

Thus, $\bar{\pi}$ is a Nash equilibrium. Lastly, this implies that the value function corresponds to a Nash value function $V^{\text{NE}}$, the return corresponds to a Nash return $J^{\text{NE}}$. $\qquad\square$

## D   CASTING HAPPO AS HAML

The maximisation objective of agent $i_m$ in HAPPO is

$$\mathbb{E}_{s \sim \rho_{\boldsymbol{\pi}_{\text{old}}}, \mathbf{a}^{i_{1:m-1}} \sim \boldsymbol{\pi}_{\text{new}}^{i_{1:m-1}}, a^{i_m} \sim \pi_{\text{old}}^{i_m}} \left[ \min \left( \text{r}(\bar{\pi}^{i_m}) A_{\boldsymbol{\pi}_{\text{old}}}^{i_{1:m}}(\text{s}, \mathbf{a}^{i_{1:m}}), \text{clip}\big(\text{r}(\bar{\pi}^{i_m}), 1 \pm \epsilon\big) A_{\boldsymbol{\pi}_{\text{old}}}^{i_{1:m}}(\text{s}, \mathbf{a}^{i_{1:m}}) \right) \right].$$

Fixing $s$ and $\boldsymbol{a}^{i_{1:m-1}}$, we can rewrite it as

$$\mathbb{E}_{\text{a}^{i_m} \sim \bar{\pi}^{i_m}} \left[ A_{\boldsymbol{\pi}_{\text{old}}}^{i_{1:m}}(\text{s}, \boldsymbol{a}^{i_{1:m-1}}, \text{a}^{i_m}) \right] - \mathbb{E}_{\text{a}^{i_m} \sim \pi_{\text{old}}^{i_m}} \left[ \text{r}(\bar{\pi}^{i_m}) A_{\boldsymbol{\pi}_{\text{old}}}^{i_{1:m}}(s, \boldsymbol{a}^{i_{1:m-1}}, \text{a}^{i_m}) \right.$$

$$\left. - \min \left( \text{r}(\bar{\pi}^{i_m}) A_{\boldsymbol{\pi}_{\text{old}}}^{i_{1:m}}(\text{s}, \boldsymbol{a}^{i_{1:m-1}}, \text{a}^{i_m}), \text{clip}\big(\text{r}(\bar{\pi}^{i_m}), 1 \pm \epsilon\big) A_{\boldsymbol{\pi}_{\text{old}}}^{i_{1:m}}(s, \boldsymbol{a}^{i_{1:m-1}}, \text{a}^{i_m}) \right) \right].$$

By the multi-agent advantage decomposition,

$$\mathbb{E}_{\text{a}^{i_m} \sim \bar{\pi}^{i_m}} \left[ A_{\boldsymbol{\pi}_{\text{old}}}^{i_{1:m}}(\text{s}, \boldsymbol{a}^{i_{1:m-1}}, \text{a}^{i_m}) \right]$$

$$= A_{\boldsymbol{\pi}_{\text{old}}}^{i_{1:m-1}}(s, \boldsymbol{a}^{i_{1:m-1}}) + \mathbb{E}_{\text{a}^{i_m} \sim \bar{\pi}^{i_m}} \left[ A_{\boldsymbol{\pi}_{\text{old}}}^{i_m}(\text{s}, \boldsymbol{a}^{i_{1:m-1}}, \text{a}^{i_m}) \right].$$

Hence, the presence of the joint advantage of agents $i_{1:m}$ is equivalent to the multi-agent advantage of $i_m$ given $\boldsymbol{a}^{i_{1:m-1}}$ that appears in HAMO. Hence, we only need to show that that the subtracted term is an HADF. Firstly, we change $\min$ into $\max$ with the identity $-\min f(x) = \max[-f(x)]$.

$$\mathbb{E}_{\text{a}^{i_m} \sim \pi_{\text{old}}^{i_m}} \left[ \text{r}(\bar{\pi}^{i_m}) A_{\boldsymbol{\pi}_{\text{old}}}^{i_{1:m}}(s, \boldsymbol{a}^{i_{1:m-1}}, \text{a}^{i_m}) \right.$$

$$\left. + \max \left( -\text{r}(\bar{\pi}^{i_m}) A_{\boldsymbol{\pi}_{\text{old}}}^{i_{1:m}}(\text{s}, \boldsymbol{a}^{i_{1:m-1}}, \text{a}^{i_m}), -\text{clip}\big(\text{r}(\bar{\pi}^{i_m}), 1 \pm \epsilon\big) A_{\boldsymbol{\pi}_{\text{old}}}^{i_{1:m}}(s, \boldsymbol{a}^{i_{1:m-1}}, \text{a}^{i_m}) \right) \right]$$

which we then simplify

$$\mathbb{E}_{\text{a}^{i_m} \sim \pi_{\text{old}}^{i_m}} \left[ \max \left( 0, \big[\text{r}(\bar{\pi}^{i_m}) - \text{clip}\big(\text{r}(\bar{\pi}^{i_m}), 1 \pm \epsilon\big)\big] A_{\boldsymbol{\pi}_{\text{old}}}^{i_{1:m}}(s, \boldsymbol{a}^{i_{1:m-1}}, \text{a}^{i_m}) \right) \right]$$

$$= \mathbb{E}_{\text{a}^{i_m} \sim \pi_{\text{old}}^{i_m}} \left[ \text{ReLU}\left( \big[\text{r}(\bar{\pi}^{i_m}) - \text{clip}\big(\text{r}(\bar{\pi}^{i_m}), 1 \pm \epsilon\big)\big] A_{\boldsymbol{\pi}_{\text{old}}}^{i_{1:m}}(s, \boldsymbol{a}^{i_{1:m-1}}, \text{a}^{i_m}) \right) \right].$$

As discussed in the main body of the paper, this is an HADF.

# E ALGORITHMS

---

**Algorithm 2:** HAA2C

---

**Input:** stepsize $\alpha$, batch size $B$, number of: agents $n$, episodes $K$, steps per episode $T$,
mini-epochs $e$;

**Initialize:** the critic network: $\phi$, the policy networks: $\{\theta^i\}_{i \in \mathcal{N}}$, replay buffer $\mathcal{B}$;

**for** $k = 0, 1, \ldots, K - 1$ **do**

    Collect a set of trajectories by letting the agents act according their policies, $\mathrm{a}^i \sim \pi^i_{\theta^i}(\cdot^i | \mathrm{o}^i)$;

    Push transitions $\{(\mathrm{o}^i_t, \mathrm{a}^i_t, \mathrm{o}^i_{t+1}, \mathrm{r}_t), \forall i \in \mathcal{N}, t \in T\}$ into $\mathcal{B}$;

    Sample a random minibatch of $B$ transitions from $\mathcal{B}$;

    Estimate the returns $R$ and the advantage function, $\hat{A}(\mathrm{s}, \mathbf{a})$, using $\hat{V}_\phi$ and GAE;

    Draw a permutation of agents $i_{1:n}$ at random;

    Set $M^{i_1}(\mathrm{s}, \mathbf{a}) = \hat{A}(\mathrm{s}, \mathbf{a})$;

    **for** *agent* $i_m = i_1, \ldots, i_n$ **do**

        Set $\pi^{i_m}_0(\mathrm{a}^{i_m} | \mathrm{o}^{i_m}) = \pi^{i_m}_{\theta^{i_m}}(\mathrm{a}^{i_m} | \mathrm{o}^{i_m})$;

        **for** *mini-epoch*$= 1, \ldots, e$ **do**

            Compute agent $i_m$'s policy gradient

$$\mathbf{g}^{i_m} = \nabla_{\theta^i} \frac{1}{B} \sum_{b=1}^{B} M^{i_m}(\mathrm{s}_b, \mathbf{a}_b) \frac{\pi^{i_m}_{\theta^{i_m}}(\mathrm{a}^{i_m}_b | \mathrm{o}^{i_m}_b)}{\pi^{i_m}_0(\mathrm{a}^{i_m}_b | \mathrm{o}^{i_m}_b)}.$$

            Update agent $i_m$'s policy by

$$\theta^{i_m} = \theta^{i_m} + \alpha \mathbf{g}^{i_m}.$$

        Compute $M^{i_{m+1}}(\mathrm{s}, \mathbf{a}) = \frac{\pi^{i_m}_{\theta^{i_m}}(\mathrm{a}^{i_m} | \mathrm{o}^{i_m})}{\pi^{i_m}_0(\mathrm{a}^{i_m} | \mathrm{o}^{i_m})} M^{i_m}(\mathrm{s}, \mathbf{a})$ //unless $m = n$;

    Update the critic by gradient descent on

$$\frac{1}{B} \sum_{\mathrm{s}} \left( \hat{V}_\phi(\mathrm{s}_b) - R_b \right)^2.$$

Discard $\phi$. Deploy $\{\theta^i\}_{i \in \mathcal{N}}$ in execution;

---

---

**Algorithm 3:** HADDPG

---

**Input:** stepsize $\alpha$, Polyak coefficient $\tau$, batch size $B$, number of: agents $n$, episodes $K$, steps per episode $T$, mini-epochs $e$;

**Initialize:** the critic networks: $\phi$ and $\phi'$ and policy networks: $\{\theta^i\}_{i \in \mathcal{N}}$, replay buffer $\mathcal{B}$, random processes $\{\mathcal{X}^i\}_{i \in \mathcal{N}}$ for exploration;

**for** $k = 0, 1, \ldots, K-1$ **do**

    Collect a set of transitions by letting the agents act according to their deterministic policies with the exploratory noise
$$\mathbf{a}^i = \mu^i_{\theta^i}(\mathbf{o}^i) + \mathcal{X}^i_t.$$

    Push transitions $\{(\mathbf{o}^i_t, \mathbf{a}^i_t, \mathbf{o}^i_{t+1}, \mathbf{r}_t), \forall i \in \mathcal{N}, t \in T\}$ into $\mathcal{B}$;

    Sample a random minibatch of $B$ transitions from $\mathcal{B}$;

    Compute the critic targets
$$y_t = \mathbf{r}_t + \gamma Q_{\phi'}(\mathbf{s}_{t+1}, \mathbf{a}_{t+1}).$$

    Update the critic by minimising the loss
$$\phi = \arg\min_\phi \frac{1}{B} \sum_t \left(y_t - Q_\phi(\mathbf{s}_t, \mathbf{a}_t)\right)^2.$$

    Draw a permutation of agents $i_{1:n}$ at random;

    **for** *agent* $i_m = i_1, \ldots, i_n$ **do**

        Update agent $i_m$ by solving
$$\theta^{i_m} = \arg\max_{\hat{\theta}^{i_m}} \frac{1}{B} \sum_t Q_\phi\left(\mathbf{s}_t, \mu^{i_{1:m-1}}_{\theta^{i_{1:m-1}}}(\mathbf{o}^{i_{1:m-1}}_t), \mu^{i_m}_{\hat{\theta}^{i_m}}(\mathbf{o}^{i_m}_t), \mathbf{a}^{i_{m+1:n}}_t\right).$$

        with $e$ mini-epochs of deterministic policy gradient ascent;

    Update the target critic network smoothly
$$\phi' = \tau\phi + (1-\tau)\phi'.$$

Discard $\phi$. Deploy $\{\theta^i\}_{i \in \mathcal{N}}$ in execution;

---

# F  EXPERIMENTS

## F.1  COMPUTE RESOURCES

For compute resources, We used one internal compute servers which consists consisting of 6x RTX 3090 cards and 112 CPUs, however each model is trained on at most 1 card.

## F.2  HYPERPARAMETERS

We implement the MAA2C and HAA2C based on HAPPO/HATRPO (Kuba et al., 2022a). We offer the hyperparameter we use for SMAC in table 1 and for Mujoco in table 2.

Table 1: Common hyperparameters used in the SMAC domain.

| hyperparameters | value | hyperparameters | value | hyperparameters | value |
|---|---|---|---|---|---|
| critic lr | 5e-4 | optimizer | Adam | stacked-frames | 1 |
| gamma | 0.99 | optim eps | $1e-5$ | batch size | 3200 |
| gain | 0.01 | hidden layer | 1 | training threads | 64 |
| actor network | mlp | num mini-batch | 1 | rollout threads | 8 |
| hypernet embed | 64 | max grad norm | 10 | episode length | 400 |
| activation | ReLU | hidden layer dim | 64 | use huber loss | True |

Table 2: Common hyperparameters used for MAA2C-NS, MAA2C-S and HAA2C in the Multi-Agent MuJoCo.

| hyperparameters | value | hyperparameters | value | hyperparameters | value |
|---|---|---|---|---|---|
| critic lr | $1e-3$ | optimizer | Adam | num mini-batch | 1 |
| gamma | 0.99 | optim eps | $1e-5$ | batch size | 4000 |
| gain | 0.01 | hidden layer | 1 | training threads | 8 |
| std y coef | 0.5 | actor network | mlp | rollout threads | 4 |
| std x coef | 1 | max grad norm | 10 | episode length | 1000 |
| activation | ReLU | hidden layer dim | 64 | eval episode | 32 |

In addition to those common hyperparameters, we set the mini-epoch for HAA2C as 5. For actor learning rate, we set it as 2e-4 for HalfCheetah and Ant while 1e-4 for Walker2d.

We implement the MADDPG and HADDPG based on the Tianshou framework (Weng et al., 2021). We offer the hyperparameter we use in table 3 and 4.

Table 3: Hyper-parameter used for MADDPG/HADDPG in the Multi-Agent MuJoCo domain

| hyperparameters | value | hyperparameters | value | hyperparameters | value |
|---|---|---|---|---|---|
| actor lr | $3e-4$ | optimizer | Adam | replay buffer size | 1e6 |
| critic lr | $1e-3$ | exploration noise | 0.1 | batch size | 1000 |
| gamma | 0.99 | step-per-epoch | 50000 | training num | 20 |
| tau | 0.1 | step-per-collector | 2000 | test num | 10 |
| start-timesteps | 25000 | update-per-step | 0.025 | epoch | 200 |
| hidden-sizes | $[64, 64]$ | episode length | 1000 | | |

Table 4: Parameter n-step used for MADDPG/HADDPG in the Multi-Agent MuJoCo

| task | value | task | value | task | value |
|---|---|---|---|---|---|
| Reacher $(2 \times 1)$ | 5 | Hopper $(3 \times 1)$ | 20 | Walker $(3 \times 2)$ | 5 |
| Ant $(4 \times 2)$ | 20 | Swimmer $(2 \times 1)$ | 5 | Humanoid $(9|8)$ | 5 |

