# OpenReview forum: "Heterogeneous-Agent Mirror Learning"
_ICLR.cc/2023/Conference — Submitted to ICLR 2023_

### Official Review · Reviewer_T26V · 2022-10-25

**Confidence:** 2
**Clarity, Quality, Novelty And Reproducibility:** This paper is of nice quality and is …
**Correctness:** 3
**Technical Novelty And Significance:** 4
**Empirical Novelty And Significance:** 4
**Recommendation:** 8

**Strength And Weaknesses:**

## Strength
- This paper is well-written and nicely organized. The proofs seem to be theoretically sound to me.
- The topic of this paper, designing good MARL algorithms that enjoys proper convergence and reward improvement property, is the core of the relevant literature and is of great significance.
- The proposed framework is novel to me and is nicely motivated. The theoretical guarantee is strong.
- The experiments are extensive and convincing.

## Weaknesses
- The maximization step of HAML seems to be the bottleneck of computational complexity. The authors remark that a few steps of gradient ascent are used in this process. The characterization of the number of steps needed is missing. It would be helpful if the authors could provide discussions on this part to complete the store of the framework.



**Summary Of The Paper:**

This paper studies MARL problems and proposes the Heterogeneous-Agent Mirror Learning (HAML) framework that enjoys monotonic improvement of the joint reward and convergence to NE. They show that HAML includes HATRPO and HAPPO as special instances. They provide HAML extensions of A2C and DDPG and show their effectiveness in experiments.

**Summary Of The Review:**

In summary, this paper is well-written and the topic is of great importance. The proposed framework is nicely motivated and enjoys a strong theoretical guarantee. The framework includes important SOTA algorithms as special instances. The extensions of the framework are achieving competitive performance in experiments.

---

> ### Author Response · Authors · 2022-11-08
> **Response to Reviewer T26V**
>
> **Q1** The authors could comment on the computational complexity of the maximization step. As it involves the sequential update and a few steps of gradient ascent, it might induce an overhead.
>
> > **A1** Another reviewer asked us about it and this is indeed a good question. We appreciate it. HAML algorithms that update agents one-by-one (in the sequential update scheme) will be slower than algorithms that update jointly, or share parameters. We meet a tradeoff: we can choose between the agents' final performance (HAML methods) or its speed (e.g., parameter-sharing methods). As authors, we do acknowledge that there will be tasks which do not require principled approaches to be solved, and thus a fast heuristic, such as MADDPG, will suffice (see Figure 3). Other tasks, however, might be harder, and a theoretically grounded appraoch will stand out (see Figures 1 & 2), although it will be slower.
> We would like to note that this concern may be alleviated in the future: recent work [Wen et al., 2022] demonstrated how to use transformes to implement sequential updates in parallel, thus speeding up the procedure.
>
> ### References
> [Wen et al., 2022] [https://arxiv.org/pdf/2205.14953.pdf]

---

### Official Review · Reviewer_iCs6 · 2022-10-27

**Confidence:** 4
**Correctness:** 3
**Technical Novelty And Significance:** 2
**Empirical Novelty And Significance:** 2
**Recommendation:** 3

**Clarity, Quality, Novelty And Reproducibility:**

Equation (1) uses the advantage with 2 arguments without explaining the intent that the implicit third argument is the (lack of) action for an empty set of agents.

While Proposition 2 is technically correct, its inclusion is misleading.  It is being used to critique agents merely finding an “improving direction.”  But the Proposition actually assumes agents all take the full step to their best response.  If the agents actually take a small enough step then it really will be an improvement even without coordination.  Perhaps there is a more subtle point to be made here about the importance of step size control, but this example is simply too blunt as a critique of standard multiagent policy gradient methods.  In particular, it is incorrect to claim without further evidence that MAA2C and MAPPO get “caught” or “fall into the trap of Proposition 2”

There lack of comparison to COMA at 3.2 seems an omission given the discussion above regarding the relative convergence guarantees.

The notation \mathbb{P} doesn’t appear to be defined.  I assume it means powerset?

The arguments do the HADF are not clearly explained.  It is ambiguous which policy is which.  The role of \mathbb{P}(-i) is not explained.  Presumably this is how we get the j on the right hand side?  But note that we need an order for j per the definition of that notation and sets are unordered.

More broadly the definition of a HADF is hard to parse.  It would help to break the giant sentence that runs through most of the definition environment into smaller, more manageable pieces.

Lemma 2 should require (5) to hold for all choices of m in addition to all choices of s.

The proof of Theorem 1 is in appendix C, not A.

In the proof of Proposition 2, the derivation before (8) is for \pi_old, not \pi_new

In the first eqnarray of the proof of Lemma 2, the expectation over actions gets lost in the middle two lines.


**Strength And Weaknesses:**

On the positive side, in combining these two ideas the paper provides additional insight into the (2022a) paper by both simplifying and generalizing the features needed and available (respectively) to guarantee policy improvement in cooperative MARL.  While this doesn’t require a lot of truly novel technical content, with the empirical demonstration of the benefits I think overall I would be happy to see these results in a good conference

On the negative side, the presentation is deficient in four respects.  First, the approach is presented as a monolithic whole, without identifying for readers some of the simplicity of the results.  In my view the whole inclusion of the drift functional / mirror learning is essentially a subsidiary result.  Theorem 1 holds even for the trivial drift functional, which means the key insight of the paper around the multi-agent advantage decomposition alone being sufficient is being buried under a lot of irrelevant technical detail.  Indeed, the HADF barely appears in the proof of Theorem 1 except as bookkeeping that could be replaced by 0.  In fact I think even greater generality is possible; the theorem requires a sequence of permutations over the agents while I believe it actually suffices that all agents be allowed to update an infinite number of times (although perhaps the permutation approach is more data efficient).  This detail is also irrelevant to the empirical results, which as the paper points out use neither HADFs nor neighborhood operators.  So while this material would be nice to include as an extension, it actually obscures and detracts from the main thrust of the paper.

Second, the paper presents as novel insights things which have long been known in the cs/econ literature on multiagent learning.  At least as far back as the 1996 Monderer and Shapley “Potential Games” paper, the convergence of dynamics where one agent at a time improves to Nash equilibrium in identical interest games has been well known.  Similarly, the problems of miscoordination when multiple agent update at the same time (which is explained in a needlessly complex way in Proposition 2) are well understood.  See, for example, Example 2 from “The Logit-Response Dynamics” by Alos-Ferrer and Netzer.  So while there is certainly some technical novelty in bringing these ideas to the richer cooperative MARL setting, the ideas themselves are not new.  Similarly, the limitations of homogeneous policies are well known.

Third, this exemplifies a broader issue where the paper overstates its significance.  This happens at a low level with word choices which are excessive such as “fundamental theorem”, “endow MARL researchers”, and .  At a higher level, it is misleading to say in the introduction that CDTE methods lack “convergence results of any kind”.  COMA certainly comes with a convergence proof.  Kuba et al. 2022a are more careful, pointing out rather that COMA doesn’t provide any trust-region-style control of the step size and so is less stable.  But while this paper enables that, it doesn’t have any theory to establish how using it guarantees better stability.  So in that sense I don’t think the convergence theory in this paper is any stronger than in COMA.  While this sort of update does empirically seem to promote stability, as already mentioned the experiments in this paper don’t use it!  The paper also fails to give due credit to the two Kuba et al. 2022 papers.  The proof of Theorem 1 is straightforward and seems largely the same as the proof of the main result from the 2022a paper, but no technical credit is given to that paper for this.  The material in 4.1 is essentially a direct generalization of the material in the 2022b paper, and Lemma 2 is a direct generalization of their Lemma 3.3.  Yet that paper is mentioned only once in passing in that part.

Fourth, there are a number of small errors, unclear notation, and issues with definitions that should be corrected.  See details below.


**Summary Of The Paper:**

Recently Kuba et al. (2022a) introduced two cooperative MARL algorithms, HATRPO and HAPPO, which employed the idea of using a version of a multiagent advantage function which captures agents updating their portion of a policy sequentially rather than simultaneously and showed that it provides a monotone policy improvement guarantee.  Also recently, Kuba et al. (2022b) introduced Mirror learning as a way of understanding why algorithms like TRPO and PPO have a monotone improvement property.  This paper combines the two ideas showing that a richer class of algorithms which have both sequential advantage-based updates and use Mirror-learning style control of updates have the monotone improvement property.  This is empirically demonstrated by showing that instances of A2C and DDPG using the sequential advantage updates outperform traditional versions.

**Summary Of The Review:**

Acceptable results, but presentation that is deficient in key ways.

---

> ### Author Response · Authors · 2022-11-08
> **Message to Reviewer iCs6**
>
> Dear Reviewer iCs6.
>
> *From your comments, we can tell that you have a deep understanding of multi-agent learning and that you have read our paper in detail. Your comments helped us correct errors in our manuscript and turn it into a better shape which we appreciate. Nevertheless, while in your ***Summary Of The Review*** you agreed on importance of our results, you still gave us a low score based on the paper's presentation. As other reviewers have different opinion on it, and we believe we have addressed all your comments; therefore, we would really appreciate it if you could read our responses and amendments and reconsider your assessment.*

---

> ### Author Response · Authors · 2022-11-08
> **Response to Reviewer iCs6 (Part 1)**
>
> **Q1** It seems to me that the introduction of such complex structures as HADF or the neighbourhood operator seems unnecessary to me. As it is implied by the theory, the results would hold even for the trivial HADF and neighbourhood---that is without imposing any of these. Could the authors explain that complexity?
> > **A1** While we cannot disagree with the technical aspect of this comment (the trivial operators do "work"), we believe that there is a slight conceptual misunderstanding in it. The ultimate aim of our theory is to improve the MARL practices. It has been empirically confirmed, already in the single-agent RL, that completely unconstrained updates are unstable due to errors in estimation of value functions and maximization oracles from samples (see, for example, the discussion in  [Schulman et al., 2017]). As a result, algorithm designers often seek for alternative methods which trade soundness for stability. Our work reveals that they **do not** have to do it. With an appropriate choice of an HADF and neighbourhood, they can choose an algorithm that provably aims to maximize the joint return, while remains stable in a given environment (the best-performing configurations may differ between tasks).
>
> **Q2** To my knowledge, the results of improvement due to sequential update have been known in the multi-agent leanring community [Monderer and Shapley, 1996]. Also, the phenomenon described in Proposition 2 (performance damage due to a joint update) has been known (Example 2 in [Alos-Ferrer and Netzer, 2010]). Could the authors comment on that?
>
> > **A2** We disagree with the first statement. To begin with, we would like to highlight that we work in the Markov game setting: we must deal with many different states and stochastic transition dynamics. This stands in contrast to games described in [Monderer and Shapley], which do not posses states. Also, their theory does not classify the set of constraints (neither soft nor hard) that would still enable convergence, like our HAML theory does. Without diving into a list of such differences, we would simply say that that paper **is not relevant**.
> When it comes to the novelty of our Proposition 2, we agree with the reviewer that it is not outstandingly innovative. We are actually glad for the provided citation---we will be happy to include it in our paper's final version. Nevertheless, we consider Proposition 2 a simple, illustrative example of the downsides of the joint update in MARL.
>
> **Q3** Aren't the statements about the *lack of convergence guarantees of any kind* exaggerated? For example, COMA has some convergence guarantees. Consequently, the statements about the paper's significance are too strong.
>
> > **A3** We believe that the context which the reviewer refers to was not articulated clearly by us. The algorithms that lack convergence guarantees are, as cited in the sentence that the reviewer referred to, MATRPO, IPPO, and MAPPO. This issue was thoroughly discussed in the (also cited) work by [Kuba et al., 2022a]. We agree with the reviewer that COMA has convergence guarantees (to a model-dependent local maximum under compatible function approximation). There is no statement made in the paper that refutes it.
> As for our choice of wording, indeed our goal is to *endow MARL researchers* with an algorithm design framework that comes with theoretical guarantees. Could the reviewer explain this phrase's incorrectness and propose an alternative? Issues with it remain a bit opaque for us so far.
>
>
> ### References
> [Schulman et al., 2022] Schulman, John, et al. "Proximal policy optimization algorithms." arXiv preprint arXiv:1707.06347 (2017).

---

> > ### Comment · Reviewer_iCs6 · 2022-12-02
> > **Regarding Q2 and Q3**
> >
> > Q2: Despite the complexity, this observation is still quite relevant.  A standard game theory trick is to turn anything into a normal form game by enlarging the set of strategies.  In this context, each deterministic policy could be considered as a separate strategy and the resulting game is a potential game and all the results apply.  As I wrote, there is still technical novelty in dealing with things like randomized policies and some of the other technical details, but credit should be given to this earlier line of work about the importance of sequential updates regardless of whether the rediscovery of the idea in this line of MARL work was totally independent.
> >
> > Q3: Endow has the connotation that what is being given is in some sense major or substantial, which is why I have the same issue with it as with describing one of the theorems as "fundamental".  They overstate the magnitude of the contribution.  A more modest version of that sentence would be something like
> >
> > "The purpose of HAML is to provide a template for rigorous algorithmic design which guarantees the method’s correctness upfront, allowing effort to be focused on aspects such as effective implementation through deep neural networks."

---

> ### Author Response · Authors · 2022-11-08
> **Response to Reviewer iCs6 (Part 2)**
>
> **Q5** Not all elements of HAML were used in the experiments.
>
> > **A5** The reviewer is right in pointing that out. Nevertheless, the key HAML concepts---the advantage decomposition, maximization oracle for every agent, and the sequential update---are tested. As we mentioned, the role of our experiments was didactic in the sense that they showed that HAML can be viewed as a *fix* of the issues of MARL. We believe to have clearly stated that there exist state-of-the-art algorithms that do utilize more HAML components (HATRPO and HAPPO). One could say that the corresponding experiments on usefulness of HAML were conducted in the paper that introduced them. Competing against these methods was not the objective of our theoretical work.
>
> **Q6** Clarity concerns and errors.
>
> > **A6** We appreciate the reviewer's careful analysis. We have revised our paper and corrected the writing errors. Also, we would like to answer reviewer's questions:
> > * The notation $\mathbb{P}$ assumes the power set. We defined it in the Problem Furmulation section.
> > * We are not sure what causes confusion in the definition of HADF. To make it clear, we wrote an intuitive explanation just below the Definition 1. Further, assuming that the reviewer has access to the version with color: the subscript joint policy (which is the "old" policy) is in black and the newly-updated other agents' policies (that we condition on) are purple. Also, while we agree with the reviewer that sets are unordered, it is not true that our definition requires knowledge of order. We simply use the ordered-tuple notation to avoid introducting extra definition. Indeed, our theoretical multi-agent RL paper, with its contents compressed to 9 pages, is complicated enough.

---

> > ### Comment · Reviewer_iCs6 · 2022-12-02
> > **Regarding Q6**
> >
> > I was indeed reading the paper printed in black and white, which makes this part hard to parse because the color coding used does not come through.  I agree that it is clearer in color.  I would suggest at least mentioning the use of color in the text so that readers of a printed version are aware to consult the color version.

---

### Official Review · Reviewer_Axzm · 2022-10-29

**Confidence:** 4
**Correctness:** 4
**Technical Novelty And Significance:** 3
**Empirical Novelty And Significance:** 2
**Recommendation:** 6

**Clarity, Quality, Novelty And Reproducibility:**

Clarity
Concepts, even the borrowed ones, are well introduced. Theorems build up in an easy flow.
Results should be clarified or normalized against any computation overhead.

Quality
Evaluation environments and tested baselines are varied. Unresolved curiosities are mostly addressed in the appendix (except the training time / computation overhead). Position in literature is clear, and contribution to better understanding the growing number of MARL algorithms is an appealing argument.

Originality
To the best of my knowledge, HAMO, HAML, and HADF are original. However, it is almost easy to misread the paper as a 'part 2' or 'chapter 2' paper to the Kuba2022b mirror learning paper. Perhaps, after establishing a clear connection to that paper initially, the rest of this HAML paper could proceed with fewer references to the mirror learning paper and instead develop a detached and independent series of claims and logical arguments.

**Strength And Weaknesses:**

Paper's main strength lies in the provability of more stable convergence properties. Even though mirror learning concepts have been proposed in closely relevant previous works, extending those results to the multi-agent setting is far from trivial. Authors make a strong case for the correctness argument of HAMO and its monotonic improvement over the course of the training iterations.

Tracing the mirror learning idea back to the Kuba2022b paper does suggest that HAMO is meant to operate with policy gradient methods only, it would be nicer if a clear delimitation/disclaimer is made in the present paper. On a related note, some comments on representative value-based methods and/or communication-based MARL methods can help better position HAML and clarify which methods are/aren't HAMO's competitors.

The theorems and proofs support just exactly what is claimed earlier in the paper, but the current version reads a little like a dry array of correctly built up arguments. Insightful elaboration and a curated interpretation of what's between and beyond the equations would be an excellent addition. For example, at the bottom of page 5, what real-world implications do these two facts carry? Does HAMO collapse to a special case?

As is the case with provably convergent MARL algorithms, the training time/computation cost may have been the factor traded off for the convergence. Figures 1 to 3 are all plotted against environment steps, but one step of training a HAML instance would carry an expensive computation (Compute the advantage function A_\pi_k in Algorithm Template 1). Please comment on the computation overhead.

**Summary Of The Paper:**

Authors present a provably convergent MARL algorithmic template HAML, from which two existing MARL algorithms are derived as demonstration. In addition, further two MARL algorithms are cast as HAML instances to rid of their shortcoming defined as 'traps' in the paper. SMAC and MAMuJoCo evaluation shows better performance, or in cases of comparable reward levels, narrower confidence intervals.

**Summary Of The Review:**

Authors make a clearly positioned contribution to explain policy gradient MARL algorithms in terms of their convergence properties and empirically show their findings. Minor points of revision include: delimitation with respect to value-based/communication-based MARL, more interpretive technicalities such as in Definition 3, and some comments on training time / computation.

---

> ### Author Response · Authors · 2022-11-08
> **Response to Reviewer Axzm**
>
> **Q1** It seems that to implement HAML in practice one must use policy-gradient methods. In MARL, there is an alternative class of value-based algorithms. Also, there is another paradigm of MARL, which is that when agents are allowed to communicate. It would be helpful if authors specified explicitly where in the literature their work should be placed: that is, MARL with decentralized (non-communicating) policies and actor-critic algorithms.
>
> > **A1** The reviewer is right in pointing out that we tackle the problem of MARL without communication and this should be highlighted. What does the reviewer think about changing the first sentence in abstract to
> *The necessity for cooperation among **independent** intelligent machines has popularised cooperative multi-agent reinforcement learning (MARL) in the artificial intelligence (AI) research community.* ?
> Further, we agree with the reviewer that HAML does not capture value-based methods, as these do not learn a policy function, but rather aim to learn the optimal value function. In HAML, the policy is a crucial object of study and thus actor-critic algorithms must be used in large-scale settings. We can offer changing a sentence in abstract to *To resolve these issues, in this paper,we introduce a novel framework named Heterogeneous-Agent Mirror Learning (HAML) that provides a general template for MARL **actor-critic** algorithms.*
>
> **Q2** Currently, the paper is a bit dry to read---it lists the mathematical results without explaining their application or intuition behind them. For example, what is the meaning of Definition 3 at the bottom of Page 5?
>
> > **A2** We understand the reviewer's concern. Indeed, our paper has a very technical form. Nevertheless, we do not agree that is devoid of explanations or interpretations. While still relating to the technical aspect of MARL, they do appear. For example, see the paragraphs under Equation (4) which explain the meaning of Propositions 1 & 2, or the discussion about different HAML components under Algorithm Template 1 on page 6. While the reviewer is right in pointing out that we did not introduce discussions on connections of HAML with the real-world applications, we consider it natural for a theoretical, algorithmic paper.
>
> **Q3** Could the authors comment on the computational overhead of HAML?
> > **A3** This is a great question. Indeed, HAML algorithms that update agents one-by-one (with steps of gradient ascent and in the sequential update scheme) will be slower than algorithms that do it jointly, or share parameters. We meet a tradeoff: we can choose between the agents' final performance (HAML methods) or its speed (e.g., parameter-sharing methods). As authors, we do acknowledge that there will be tasks which do not require principled approaches to be solved, and thus a fast heuristic, such as MADDPG, will suffice (see Figure 3). Other tasks, however, might be harder, and a theoretically grounded appraoch will stand out (see Figures 1 & 2), although it will be slower.
> Lastly, we would like to note that this concern may be alleviated in the future: recent work [Wen et al., 2022] demonstrated how to use transformes to implement sequential updates in parallel, thus speeding up the procedure.
>
> ### References
> [Wen et al., 2022] [https://arxiv.org/pdf/2205.14953.pdf]

---

### Official Review · Reviewer_iVJ4 · 2022-10-29

**Confidence:** 3
**Clarity, Quality, Novelty And Reproducibility:** The paper is ok to read, while explic…
**Correctness:** 3
**Technical Novelty And Significance:** 3
**Empirical Novelty And Significance:** 2
**Recommendation:** 6

**Strength And Weaknesses:**

+: The paper has nice result for the properties of a class of policies.

-: The main issue is that achieving NE for cooperative games may not be the best strategy. We ideally want the cooperative equilibrium in a decentralized fashion. This aspect should be detailed better.

-: The other issue is the comparisons. The entire area of CTDE is not considered for evaluations. See for instance the comparison strategies in recent NeurIPS paper: "PAC: Assisted Value Factorisation with Counterfactual Predictions in Multi-Agent Reinforcement Learning". Many of them are at https://github.com/hijkzzz/pymarl2 .

**Summary Of The Paper:**

This paper provides a general template for MARL algorithmic designs. They prove that algorithms derived from the HAML template satisfy the desired properties of the monotonic improvement of the joint reward and the convergence to Nash equilibrium.

**Summary Of The Review:**

This paper provides a general template for MARL algorithmic designs. They prove that algorithms derived from the HAML template satisfy the desired properties of the monotonic improvement of the joint reward and the convergence to Nash equilibrium. Expanding the reason for limiting to Nash equilibrium and better comparisons would have been good.

---

> ### Author Response · Authors · 2022-11-08
> **Response to Reviewer iVJ4**
>
> **Q1** In cooperative games, a Nash-equilibrium (NE) strategy may not be optimal. NEs are considered a natural solution concept for decentralized games.
> > **A1** We agree with the reviewer thst in the cooperative setting, there exists an optimal joint policy (i.e., Pareto optimality), with respect to which NEs are weaker. Crucially, however, while the setting that we tackle is cooperative, the policies of agents are indeed decentralized. In this setting finding a Pareto-optimal policy may simply be impossible, as it may involve joint exploration and joint optimization---an impossibility for agents that do not know other agents' policies.
> > As an illustrative example, consider a 2-player cooperative game without states, with a (large) payoff matrix given by
> $\begin{bmatrix}5 \ 0 \dots 0\ \newline
> 0 \ \dots 0\ \newline
> \dots\ \newline
> 0 \ \dots 10\end{bmatrix}$
> That is, there is only one joint action when agents receive a payoff of 5, one for which they receive 10, and a multitude of actions with 0 payoff. Of course, the Pareto-optimal policy is to take the joint action from the lower-right corner. Suppose, however, that the agents have learned (so far) to take the top-left action which corresponds to payoff 5. Now, whenever any of the agents explores a different action, the joint return decreases. Thus, the top-left action is an equilibrium, and without communicating their exploration, the agents have no protocol to find the optimal alternative in a large space of joint actions. Hence, the 5 is already a great achievement.
>
> **Q2** The comparisons could be more extensive. The entire range of state-of-the-art methods is not evaluated.
> > **A2** We agree with the reviewer that we have not compared against all SOTA for multi-agent RL (MARL), as opposed to how it was done in the PAC paper [Zhou et al., 2022]. However, we **do not** consider this a limitation. Unlike PAC, our paper does not propose a new SOTA deep MARL algorithm. Instead, we introduce a theory that enables design of practical algorithms with guarantees. Our experiments were supposed to verify correctness of our theory and demonstrate benefits of enhancing existing methods (e.g., MAA2C and MADDPG) with it. See the bottom paragraph of Page 7 for discussion. Our empirical results confirm that the HAML theory is correct, as well as show that simple ideas, such as MAA2C, can often benefit greatly from alignment with HAML (see Figure 2).

---

### Author Response · Authors · 2022-11-08
**Words of appreciation**

We thank all reviewers for their efforts in reading our paper. Your comments assure us that this has been done diligently. The provided feedback will help us improve the quality of our article.

Authors

---

### Decision · Program_Chairs · 2023-01-20

**Decision:**

Reject

**Justification For Why Not Higher Score:**

The reviewers identifies a number of issues while the authors did not adequately address them. The biggest concern is the presentation of the paper. The current version does not seem to be in a good shape to be published in ICLR.

**Justification For Why Not Lower Score:**

N/A

**Metareview: Summary, Strengths And Weaknesses:**

Summary:
This paper combines ideas from two recent paper to show that a richer class of algorithms which have both sequential advantage-based updates and use Mirror-learning style control of updates have the monotone improvement property in MARL. This is empirically demonstrated by showing that instances of A2C and DDPG using the sequential advantage updates outperform traditional versions.

Strength:
- the proposed algorithm may have more stable convergence properties.
-  simplifying and generalizing existing ideas
-  empirical demonstration of the benefits

Weakness:
- The presentation makes the paper hard to read
- Novelty: ideas might be known in the literature long time ago